# Neurofibromatosis Type 1 and the Search for Effective Tumor Therapies Using High-Throughput Drug Screening

**DOI:** 10.3390/curroncol32110649

**Published:** 2025-11-20

**Authors:** Stephanie J. Bouley, Benjamin E. Housden, James A. Walker

**Affiliations:** 1Center for Genomic Medicine, Massachusetts General Hospital, Boston, MA 02114, USA; sbouley@mgh.harvard.edu; 2Living Systems Institute, University of Exeter, Exeter EX4 4QD, UK; b.housden@exeter.ac.uk; 3Department of Clinical and Biomedical Science, University of Exeter, Exeter EX4 4QJ, UK; 4Cancer Program, Broad Institute of MIT and Harvard, Cambridge, MA 02142, USA; 5Department of Neurology, Massachusetts General Hospital, Harvard Medical School, Boston, MA 02114, USA

**Keywords:** neurofibromatosis type 1, high-throughput screening, drug screening

## Abstract

In this review, we examine five decades of drug screening efforts aimed at developing treatments for Neurofibromatosis Type 1 (NF1). We begin with an overview of the clinical manifestations of NF1, followed by a discussion of early attempts to target the RAS pathway before the advent of high-throughput screening technologies. Next, we describe the *in vitro* and *in vivo* models employed in these studies and summarize the major screening efforts conducted across various NF1-related cell types, including any subsequent validation or follow-up studies. We conclude with a brief assessment of how high-throughput screening has informed clinical trials and discuss future directions for therapeutic discovery in NF1.

## 1. Introduction

### 1.1. Neurofibromatosis Type 1

Neurofibromatosis type 1 (NF1) is a genetic tumor-predisposition disorder affecting about 1 in 2500 births [1]. First described in the 1800s by Dr. Friedrich von Recklinghausen, NF1 presents with a broad range of symptoms due to loss of neurofibromin, a protein encoded by the *NF1* gene, involving multiple organ systems and cell types. These include cutaneous manifestations such as café-au-lait macules and axillary or inguinal freckling, neurological issues including learning disabilities and behavioral challenges, cardiovascular and skeletal abnormalities. One of the defining, and more serious, aspects of NF1 is a predisposition to benign and malignant tumors [2,3]. Neurofibromas are the most common of these and arise from nerve sheaths—either as cutaneous neurofibromas (cNFs) on the skin, or plexiform neurofibromas (pNFs), which are larger, more complex tumors. While NF1 patients have systemic loss of one *NF1* allele due to a germline mutation, it is the acquisition of a second, somatic mutation in the remaining wild-type *NF1* gene, specifically in Schwann cells, that gives rise to neurofibromas [4]. While benign, pNFs can become large and debilitating, impairing bodily functions and drastically lowering the quality of life. Due to their invasive growth, the surgical removal of pNFs can be challenging, and, furthermore, about 15% of these benign tumors subsequently undergo transformation into malignant peripheral nerve sheath tumors (MPNSTs) [5,6]. The malignant transformation of pNFs occurs through intermediate lesions known as atypical neurofibromas or atypical neurofibromatous neoplasms of uncertain biological potential (ANNUBPs). These ANNUBPs exhibit increased cellular atypia and proliferation compared to benign NFs but lack the full malignancy of MPNSTs [7,8]. Progression from pNFs to ANNUBPs and ultimately to MPNSTs involves the accumulation of additional genetic alterations (both mutations and copy number changes) including the cell cycle regulator *CDKN2A/B*, the tumor suppressor *TP53* and components of the PRC2 complex, such as *EED* and *SUZ12*, which drive tumor aggressiveness and malignancy [9,10,11,12,13].

MPNSTs are aggressive sarcomas with very poor response to conventional chemotherapy and radiotherapy, resulting in poor prognoses, with 5-year survival rates ranging from 16% to 62% [14]. Beyond MPNSTs, individuals with NF1 carry a higher-than-average risk for various other cancers, including leukemia, breast cancer, gastrointestinal stromal tumors (GISTs), and pheochromocytomas, resulting in a reduced life expectancy—by an estimated 10 to 20 years compared to the general population—making NF1 a complex and lifelong clinical challenge [15,16].

Alongside nerve sheath tumors, NF1 individuals are also at risk for developing various low- and high-grade gliomas [17]. Low-grade gliomas (LGGs) are the most common central nervous system (CNS) tumors diagnosed in NF1 patients, occurring in ~20% of patients; this is significantly higher than patients diagnosed with high-grade gliomas (HGGs), which only occur in ~2.5% of NF1 patients [18,19]. Optic pathway gliomas (OPGs) represent about one-third of all NF1-associated LGGs [19] and affect an estimated 15–20% of children [20]. While most NF1-related OPGs remain asymptomatic, vision is affected in about 40% of affected patients. The location and infiltrative nature of these tumors mean that complete surgical resection is rarely feasible without significant risk, and effective treatment options remain limited for OPGs, LGGs, and HGGs in NF1 patients [21].

The NF1 tumor types that have received the most attention by researchers are pNFs and MPNSTs. The high prevalence and debilitating symptoms of pNFs, such as pain and nerve dysfunction, even in their benign form [22], make them priority candidates for therapeutic intervention. Successful treatment of pNFs might alleviate these symptoms along with potentially reducing the risk of malignant transformation into the most aggressive and deadly NF1-related tumor, MPNSTs.

### 1.2. Neurofibromin Functions to Regulate the RAS Signaling Pathway

Throughout the 19th and 20th centuries, physicians documented the clinical features now recognized as classic manifestations of NF1. The identification of the *NF1* gene in 1987 and its subsequent cloning led to the elucidation of the molecular function of the encoded protein, neurofibromin, in 1990, allowing the design of rational therapeutic strategies (Figure 1) [23,24,25,26,27,28]. The discovery that neurofibromin acts as a negative regulator of the RAS/MAPK signaling pathway explained how its loss leads to hyperactivation of this pathway [29], predisposing individuals to both benign and malignant tumors, and highlighted components of the pathway as promising therapeutic targets [30].

The only two FDA-approved treatments currently available for NF1 patients—both targeting MEK signaling—were approved in the USA within the last five years, marking a 30-year gap between the identification of MEK as a therapeutic target and the eventual approval of effective drugs [31,32,33]. During this time, testing individual compounds, although systematic and logical, has proven to be a slow and often unproductive process, especially when candidates fail in preclinical testing or clinical trials—as was the case for most targets explored in the 1990s, apart from MEK. While MEK inhibitors are currently only approved for treatment of pNFs, there are no effective therapies for MPNSTs.

### 1.3. Drug Screening for New Therapeutic Targets for NF1 Tumors

By the mid-2000s, high-throughput screening (HTS) and targeted drug screening approaches were gaining traction in oncology research. HTS of drugs involves rapidly testing hundreds to millions of compounds simultaneously using automated systems to identify potential candidates for treating a disease. HTS enables faster and objective discovery of promising drugs by quickly narrowing down large libraries of molecules, while individual testing allows for more detailed and focused analysis of a single drug’s effects.

There are also specialized screening strategies that are often implemented when focusing on a specific target or pathway. Two that are often mistaken for each other are *target-based screening* and *targeted-based screening*; while similar in name, these types of screens differ significantly. In *target-based screening* (or target-based drug discovery), a specific target implicated in disease progression is probed with a library of experimental small molecules or compounds designed explicitly against the target’s structure in the hopes of identifying compounds that bind and modulate that target’s activity.

In contrast, *targeted-based drug screening*, while also high-throughput, takes advantage of existing inhibitors against a specific target/pathway or inhibitors with a common status (e.g., FDA-approved or actively being tested in the clinic), to perform a more refined screen that will identify existing compounds that illicit a phenotypic response in disease-relevant cells. This allows researchers who have identified a clinically relevant target through other experimental designs, such as omics analyses, to follow-up on those targets using existing compounds. When using existing, FDA-approved therapies, this type of targeted-based drug screening is known as *drug repurposing*. Drug repurposing involves the use of already approved drugs, having already undergone safety and dosing studies in humans. This helps to reduce the risk, cost, and time associated with drug development and translation into the clinic [34].

The first results from HTS in NF1 models emerged in the early 2010s (Table 1) [35]. Since then, numerous screens, including high-throughput and more targeted lower-throughput approaches, have been conducted across a variety of NF1 models. These efforts have included small-molecule libraries, pathway-specific inhibitor screens, synthetic lethal screens, and genetic screens (Figure 2). In this review, we provide a detailed overview of these screening strategies, highlight key findings across model systems, and discuss their implications for advancing NF1 patient treatment.

### 1.4. Repurposing Existing Drugs for Treating NF1-Deficient Tumors

Drugs effective against tumors with similar molecular features, e.g., RAS pathway hyperactivity, provide a strong rationale for testing in *NF1*-deficient tumors. Drug repurposing libraries vary widely in scale and composition, encompassing multiple categories of compounds. Some, such as the National Center for Advancing Translational Sciences (NCATS) Pharmaceutical Collection, include only FDA-approved drugs or those approved by equivalent regulatory agencies in other countries. Others, like the NIH Clinical Collection, are more investigational in nature and focus on compounds with prior use in human clinical trials. Still others—such as the NCATS Mechanism Interrogation PlatEs (MIPE) library—contain a mix of approved drugs, clinical trial candidates, and preclinical compounds. Additionally, several commercial providers, including Prestwick Chemical and SelleckChem, offer customizable libraries tailored to specific research interests. In this review, we indicate which library was used where applicable.

## 2. In Vitro and In Vivo NF1 Models for High-Throughput and Targeted Drug Screening

### 2.1. In Vitro 2D NF1 Immortalized Cell Culture Models

The availability of scalable, low-cost cell models that accurately recapitulate disease mechanisms is essential for drug screening. The most fundamental of these models are cell lines derived directly from patients. Over the past 30 years, numerous *NF1*-deficient cell lines have been established from patient biopsies and tissue donations (Table 2). The earliest of these were derived from MPNSTs [39,40]. The development of patient-derived Schwann cell lines from benign pNFs has required immortalization through the introduction of human telomerase reverse transcriptase (hTERT) or SV40 large T-antigen to enable long-term culture [41,42]. A significant challenge has been the lack of appropriate control cell lines needed to distinguish candidate therapies that selectively target *NF1*-deficient tumor cells without causing general cytotoxicity. In some cases, both *NF1* heterozygous and homozygous cell lines have been established from the same pNF patient biopsy (ipNFs), providing genetically matched control cells differing only in *NF1* mutation status [41]. This pairing allows for more accurate identification of therapies with pNF-specific effects.

More recently, the same immortalization strategies of employing hTERT and CDK4 were applied to cells from cNFs, resulting in the development of new laboratory models for drug testing (icNFs) [43]. As with the pNF-derived lines, both *NF1* homozygous and heterozygous cNF cell lines have been successfully immortalized, supporting robust and controlled compound library screening efforts in this context.

Large-scale drug screening using MPNST models sometimes incorporates sporadic MPNST cell lines (i.e., those not derived from an NF1 individual) as comparators. The rationale being that if an *NF1*-deficient MPNST responds to a particular inhibitor while a sporadic MPNST does not, the effect may be attributed specifically to *NF1* loss, rather than to features common to all MPNSTs. However, genetic context plays a critical role in modulating the penetrance of responses to genetic or pharmacologic perturbations, potentially leading to variable outcomes across different models [44,45]. Using a panel of both NF1-associated and sporadic MPNST models may be necessary to confidently identify compounds that exert NF1-specific effects.

A key limitation of immortalized cell line models is lack of tumor heterogeneity. NF1-associated tumors, including pNFs, cNFs, and MPNSTs, consist of multiple interacting cell types, and single-cell-type models fail to recapitulate the *in vivo* tumor microenvironment (Figure 3A). Co-culture systems or conditioned media are two strategies to study the influence of the tumor microenvironment on *NF1*-deficient cells. For example, by co-culturing immortalized *NF1*-null (*NF1^−/−^*) pNF spheroids with primary *NF1*-heterozygous (*NF1^+/−^*) fibroblasts from pNF patients, or exposing the spheroids to fibroblast-conditioned media, pNF growth and invasiveness can be significantly enhanced compared to monocultured spheroids [46]. This highlights not only the critical role of the microenvironment in tumor progression, but also its potential as a therapeutic target to reduce pNF tumor burden.

### 2.2. Induced Pluripotent Stem Cells (iPSCs) and Primary NF1 Cell Models

An alternative to immortalizing cell lines for studying specific cell types is the derivation of patient-derived stem cells, which can subsequently be differentiated into the desired disease-relevant cell types. Two main types of stem cells can be used for this purpose: induced pluripotent stem cells (iPSCs)—produced by genetic reprogramming of adult somatic cells, such as skin or blood cells, and embryonic stem cells (ESCs)—derived from blastocysts (Figure 3B). Whereas use of human ESCs raises ethical concerns, iPSC collection is not typically contentious, as it involves direct patient or caregiver consent and reflects a personal investment in the research [58,59].

### 2.3. Isogenic NF1 Cell Culture Model Generation by CRISPR-Cas9 Technology

CRISPR-Cas9 gene editing allows the creation of isogenic *NF1*-KO cell lines that are of great utility for application in HTS studies focused on understanding how loss of the second *NF1* allele influences drug sensitivity or resistance. The use of cell lines derived from a common parental background reduces variability caused by differences in genetic background, epigenetic states, or additional mutations that may be present in unrelated cell lines. CRISPR gene editing strategies can introduce *NF1* mutations on the wild-type (WT) allele or correct mutant alleles to generate isogenic control lines. Several drug discovery screening studies have utilized CRISPR-engineered cell lines, including those derived from wild-type Schwann cells, epithelial-like cells, and iPSCs [47,48,49,50,60].

### 2.4. In Vitro Primary NF1 Cell Culture Models

In addition to the development of immortalized cell lines, researchers have also leveraged primary cells derived from patient biopsies or animal models lacking *NF1* (Figure 3C). Use of primary cell cultures presents both advantages and limitations, as with any experimental model [51]. Primary cells are generally considered more biologically relevant than immortalized cell lines since they retain a heterogeneous population and preserve natural cellular crosstalk and behavior. It is also often possible to generate primary cultures containing multiple cell types, offering a more accurate representation of the tissue of origin. Drawbacks of primary cultures include their limited lifespan, challenges in culture maintenance, variability in cell types from different isolates, and difficulties in obtaining samples from patients. The limited availability of primary cells has significantly constrained the scale of drug discovery screening efforts. Additionally, the lack of appropriate control cells can complicate data interpretation.

### 2.5. In Vitro 3D NF1 Cell Culture Models

Certain cell culture systems, including iPSCs, primary cells, and some immortalized cell lines, have the capacity to form three-dimensional (3D) spheroid or organoid models. Spheroids typically arise from the spontaneous aggregation of single-cell suspension cultures. Organoids are derived specifically from iPSCs or ESCs and are designed to recapitulate the architecture and cellular composition of their tissue of origin [52]. These 3D models offer significant advantages for therapeutic research, as they more accurately mimic *in vivo* structures and tumor-stroma interactions [53]. While NF1-specific spheroid and organoid models are currently limited, several have been developed and successfully applied in screening studies. Both cNF and pNF organoid models have been shown to retain cellular compositions closely resembling those of patient biopsy samples, making them particularly valuable for drug screening applications [46,61].

### 2.6. Single-Celled Organisms and Cell Lines Derived from NF1 Invertebrate Models

Non-human NF1 cell models have proven valuable for drug screening efforts. Both yeast and *Drosophila* have been employed in synthetic lethal screening approaches (Figure 3D), offering rapid and cost-effective platforms for large-scale screening. *Saccharomyces cerevisiae* (budding yeast) has been used in at least one HTS to identify small molecules that selectively kill yeast deficient in *ira2*, one of the two yeast orthologs of *NF1*, with the aim of developing these compounds into potential therapeutics [54]. Additionally, HTS in yeast has generated a comprehensive database of genetic interactions covering nearly all genes in the yeast genome. This resource can be mined to identify candidate therapeutic targets for genetic diseases, including *NF1* [62].

*Drosophila melanogaster* (fruit fly) models have provided a useful genetic system to investigate the neurological outcomes of *NF1* [63]. A recent study has also made use of cell lines derived from *Drosophila* in a genome-wide RNAi library synthetic lethal screen to identify novel potential therapeutic targets for NF1-associated tumors [50].

### 2.7. In Vivo Vertebrate Models of NF1 for Testing Therapeutics for NF1-Associated Tumors

Preclinical models of NF1 have been developed across three different vertebrate species—zebrafish, mice, and pig—that recapitulate neurofibromas and other tumors associated with *NF1* loss. While these are not necessarily ideal models for HTS for drug discovery, they are critical as preclinical tumor models to validate drugs identified from *in vitro* screens before potential testing in the clinic.

Genetically engineered mouse models (GEMMs) are the most extensively used *in vivo* models of NF1. These can be used to model various NF1-associated phenotypes, including tumors, skeletal defects, behavioral changes, pain, and cognitive impairments. However, different GEMMs are tailored to specific NF1 symptoms, as no single genetic background reproduces the full spectrum of *Nf1* phenotypes. A key limitation of mouse models is that *Nf1* heterozygous animals do not spontaneously acquire second-hit mutations, and global homozygous *Nf1* loss is embryonically lethal [55]. Consequently, researchers use Cre-LoxP technology to conditionally knock out (KO) *Nf1* in specific cell types to initiate tumor formation. *Nf1^fl/fl^*; *DhhCre* mice with *Nf1* KO targeted to Schwann cell precursors develop pNFs [55]. A murine model engineered with targeted *Nf1* KO in neural crest-derived cells (*Hoxb7-Cre*; *Nf1^fl/fl^*) robustly develops cNFs [63], while optic glioma formation occurs in *Nf1^fl/fl^*; *GFAP-Cre* mice [56]. *Nf1* and *p53* double mutant mice (*Nf1^+/−^*; *p53^+/−^*) have been used to establish a GEMM for *MPNSTs* [64]. In addition to providing *in vivo* NF1 tumor models for preclinical testing, cell lines have been established from several of these GEMMs, which have been useful in HTS (Figure 3D; Table 2).

Accurate modeling of NF1-associated tumors preserving the tumor microenvironment and immune contexture is critical for preclinical drug evaluation. Patient-derived xenograft (PDX) models have been developed to better emulate human MPNST responses to treatment [65]. While these models enhance translational relevance, they lack an intact immune system, a key component in tumor progression. Cell lines derived from these PDXs have also been generated and allow for *in vitro-in vivo* comparisons [66]. Ultimately, while no single mouse model recapitulates all aspects of NF1, several robust models exist for studying specific symptoms and tumor types.

The zebrafish (*Danio rerio*) has also been developed as an NF1 model. Zebrafish possess two *NF1* orthologs, *nf1a* and *nf1b*; the loss of both alleles via targeted mutagenesis results in larval lethality 7–10 days post-fertilization [67]. Before death, mutant larvae exhibit multiple phenotypes relevant to NF1, including Schwann cell hyperplasia, myelination defects, motor and learning impairments, melanization abnormalities, and hyperactive RAS signaling. However, because these larvae do not survive to adulthood, benign tumor development cannot be studied in this background—a limitation due to global rather than conditional *nf1a/b* KO. Zebrafish with combined conditionally knocked out *nf1a/b* and *p53* in neural crest cells develop gliomas and MPNSTs that resemble the histopathology of human MPNSTs. Treatment with sunitinib, a receptor tyrosine kinase inhibitor, alone or combined with MEK inhibition, was shown to reduce tumor burden in this model [68]. Additionally, this system has been used to test DNA topoisomerase I inhibitors. Combined with the scalability and low cost of zebrafish, these findings suggest that this vertebrate model may prove valuable for future automated drug screening efforts in NF1 [69].

An NF1 minipig model has been developed to address the limitations of rodent models in recapitulating the broader clinical phenotypes of NF1 [70,71]. Minipigs exhibit the hallmark features of NF1 seen in patients, including café-au-lait macules, freckling, Lisch nodules, hypopigmentation, and tibial dysplasia. Importantly, aged minipigs can also develop neurofibromas and OPGs. Both melanocytes and Schwann cells from this model undergo spontaneous loss of heterozygosity (LOH), closely mirroring the genetic events leading to tumorigenesis seen in human NF1. Furthermore, the model responds to MEK inhibition with a reduction in RAS signaling, consistent with therapeutic responses observed in patients [70].

Overall, the NF1 minipig has the potential for a highly translational preclinical model, offering a unique platform for validating candidate therapeutics identified through drug discovery screens prior to clinical testing. While no animal model fully replicates all aspects of NF1, these species-specific systems together provide a comprehensive toolkit for studying NF1 pathobiology and advancing targeted drug development.

## 3. Targeting the RAS Pathway in NF1

### 3.1. Understanding the Role of NF1 in RAS Regulation

Neurofibromin functions as a RAS GTPase-activating protein (RAS-GAP), negatively regulating the activity of RAS—a family of binary molecular switches that cycle between an active, guanosine triphosphate (GTP)-bound state and an inactive, guanosine diphosphate (GDP)-bound state [72]. Although RAS proteins possess intrinsic guanine nucleotide exchange and GTP hydrolysis capabilities, these are insufficient for the rapid cycling required for proper signal transduction [30]. To facilitate this process, RAS proteins rely on guanine nucleotide exchange factors (GEFs) to promote GDP-GTP exchange and on GAPs to accelerate GTP hydrolysis, thereby regulating RAS activity (Figure 4A). While multiple RAS-GEFs can promote GDP-to-GTP exchange, relatively few RAS-GAPs exist. Neurofibromin serves as a major negative regulator of all three classical RAS isoforms (HRAS, NRAS, and KRAS), owing to its ubiquitous expression [73]. It should be noted that the functional core of neurofibromin, the GAP-related domain (GRD), which is sufficient for its GAP activity, represents only about 10% of the protein. While nearly 6000 pathogenic or likely pathogenic *NF1* variants have been identified to date [74], most lie outside this domain. This reflects the absence of clinically relevant mutational “hotspots” in *NF1* but also hints at other functional domains beyond the GRD [75].

The RAS family plays a central role in regulating key signaling cascades, including the RAF/MEK/ERK and PI3K/AKT/mTOR pathways, which promote cell proliferation, growth, and survival. Under normal conditions, feedback mechanisms regulate RAS activation by inhibiting RAS-GEFs. For example, activated ERK can phosphorylate upstream regulators such as the GEF, Son of Sevenless (SOS), reducing GDP-to-GTP exchange on RAS [76]. However, diminished RAS-GEF activity alone does not enhance RAS-GAP function; rather, it halts further activation of RAS. It is the activity of RAS-GAPs that ultimately downregulates RAS signaling.

Loss of neurofibromin function leads to elevated RAS-GTP levels, disrupting the balance between active and inactive RAS, and resulting in elevated downstream signaling (Figure 4B). This dysregulation results in uncontrolled cell proliferation, resistance to apoptosis, and increased risk of tumor development [77]. Four primary therapeutic strategies have been explored to restore RAS pathway homeostasis: (1) blocking RAS membrane localization, (2) directly targeting RAS with small molecules, (3) inhibiting downstream RAS signaling components, and (4) restoring WT neurofibromin expression (Figure 4C).

### 3.2. RAS Inhibition

The earliest candidates investigated for their ability to inhibit RAS signaling included farnesyl transferase inhibitors (FTIs) [78], MEK inhibitors [79,80], EGFR inhibitors, and PI3K inhibitors [81]. FTIs were developed to block the addition of a farnesyl lipid group to the C-terminus of RAS, a posttranslational modification essential for its localization to the plasma membrane [82]. In the context of *NF1* loss, FTIs initially demonstrated promise by inhibiting MPNST cell line growth [78]. However, subsequent clinical trials revealed limited efficacy since FTIs failed to extend time-to-progression in children and young adults with NF1-associated progressive pNFs [78,83]. FTIs have also been evaluated in combination with lovastatin, a cholesterol-lowering agent; this combination induced apoptotic cell death in MPNST models both *in vitro* and *in vivo*, though it does not appear to have advanced to clinical trials [84,85]. Geranylgeranyl transferase inhibitors were explored as alternative prenylation inhibitors in MPNSTs, but these efforts have remained confined to preclinical studies [86]. Although RAS proteins have long been considered “undruggable,” recent advances have led to the development of small-molecule inhibitors that target specific binding pockets on the RAS protein surface. Notably, the pan-RAS inhibitor RMC-977 has emerged as a promising therapeutic candidate for NF1-related tumors, showing efficacy in both *in vitro* and *in vivo* MPNST models [87].

### 3.3. Targeting Downstream RAS Effector Proteins

MEK inhibitors initially emerged as potential anti-cancer agents in the mid-1990s, with several MEK1 and MEK1/2 inhibitors entering clinical trials for RAS-driven cancers in the early 2010s [88,89]. The first FDA approval of MEK inhibitors came in 2013 with trametinib used for the treatment of unresectable or metastatic BRAF-mutant melanoma [90]. This was followed by cobimetinib in 2015 (also for BRAF-mutant melanoma), selumetinib in 2016 for advanced thyroid cancer, and binimetinib in 2018 [91]. Given the established link between RAS hyperactivation and increased RAF/MEK/ERK signaling, MEK inhibition was a logical therapeutic avenue for testing in *NF1*-deficient cells.

Preclinical studies demonstrated that the MEK inhibitor mirdametinib reduced proliferation in both neurofibromas and MPNSTs *in vivo* mouse models (neurofibromas: *Nf1^fl/fl^; Dhh-Cre* model, MPNSTs: xenograft model), prolonged survival in mice engrafted with human MPNST cells, and led to tumor shrinkage in neurofibroma mouse models [92]. These findings laid the groundwork for initiating a clinical trial in NF1 patients. The resulting SPRINT trial demonstrated that selumetinib produced a partial response in 68% of NF1 patients with inoperable pNFs, with 82% of those responses lasting at least one year [31]. Based on these results, selumetinib received FDA approval in 2020 for the treatment of inoperable pNFs in NF1 patients aged 2–18 years.

As of 2025, only one additional MEK inhibitor has received FDA approval for treating NF1-associated pNFs: mirdametinib, the same compound used in earlier preclinical studies [33]. Unlike selumetinib, mirdametinib is approved for individuals over the age of eighteen. Meanwhile, recent clinical data confirm that selumetinib is also effective in NF1 adults with inoperable pNFs [93], expanding its potential utility beyond pediatric populations.

Beyond oral MEK inhibitors, NFlection Therapeutics has developed a topical MEK inhibitor, NFX-179. This compound is designed for local metabolic absorption and breakdown when applied to cutaneous tumor lesions [94]. A Phase 2a clinical trial in NF1 patients with cNFs demonstrated that NFX-179 significantly reduced tumor development [95]. NFX-179 currently holds orphan drug designation for the treatment of cutaneous NF1 and is expected to enter Phase 3 trials soon.

Neurofibromin was identified as a regulator of mTOR signaling in the mid-2000s, based on observations of elevated activated S6 levels in NF1 patient-derived cell lines, *NF1*-mutant optic nerve glioma mouse models, and human pilocytic astrocytoma tumors [96]. *NF1*-deficient Schwann cells were also found to be highly sensitive to the mTOR inhibitor, rapamycin [96,97]. Inhibition of PI3K, which acts upstream of mTOR, was also shown to significantly reduce glioma proliferation and optic nerve volume in NF1 mouse models [98]. Building on these *in vitro* and *in vivo* findings, at least two clinical trials have explored the use of mTOR inhibitors as monotherapy for treating NF1-associated pNFs and pediatric LGGs [99,100]. While mTOR inhibition did not lead to reductions in pNF tumor volume, patients with NF1-associated LGGs experienced either tumor shrinkage or stabilization, with minimal side effects. These results may suggest that, in certain cases, mTOR inhibition alone may be sufficient to manage specific NF1 manifestations. As discussed later, combination therapies targeting both mTOR and MEK pathways are now being investigated for some NF1-associated tumor types [101].

## 4. Screening for Novel Treatment Options for Neurofibromas

While both pNFs and cNFs arise from *NF1*-deficient Schwann cells, researchers have adopted distinct strategies for identifying novel therapeutic targets for each tumor type, owing to their developmental and biological differences. In this section, we outline HTS methodologies tailored to each tumor type.

### 4.1. Plexiform Neurofibromas

The development of a panel of patient-matched, immortalized Schwann cell lines derived from pNFs (ipNFs) [41] was a pivotal step in enabling large-scale drug screening within an NF1-relevant cellular context. These cell lines were subsequently characterized through comprehensive multi-omics analyses [42]. An initial HTS involving 1912 oncology-focused compounds—drawn from the MIPE 4.0 library—was performed across *NF1*-null (*NF1^−/−^*), wild-type (*NF1^+/+^*), and heterozygous (*NF1^+/−^*) Schwann cell lines [42]. The MIPE 4.0 library consists of 40% FDA-approved drugs, 20% investigational drugs in clinical trials, and 40% preclinical compounds, representing a broad range of mechanisms of action. This screen provided essential growth rate data for the cell lines, along with quality control metrics for drug response, exemplified by the proteasome inhibitor bortezomib. However, comprehensive discussion of the full screening results was limited in the initial report.

Follow-up analyses aimed at refining drug selection and prioritizing candidate compounds for pNF therapy were subsequently conducted [102]. These efforts integrated the previous compound library screening data [42] with transcriptomic profiles from resected human pNF tumors using a gene regulatory network-based framework. The resulting analysis identified clusters of candidate drugs ranked by therapeutic potential. The top 15 compounds from each cluster were prioritized according to computational analysis for future preclinical and clinical validation, including combinational screens.

To better replicate the *in vivo* tumor microenvironment, 3D culture models have been developed using the pNF-derived and control lines from the Wallace lab [41,103]. Both 2D and 3D culture system models were used to test the efficacy of three candidate drugs—selumetinib, picropodophyllin (an IGF-1R inhibitor), and LDN-193189 (a BMP2 inhibitor). Of the three, LDN-193189 demonstrated the greatest efficacy in inhibiting cell proliferation, supporting the rationale for further *in vivo* studies to assess BMP2 inhibition as a therapeutic strategy in *NF1*-deficient pNFs.

While most large-scale drug screening studies focus on *in vitro* drug potency metrics such as the half-maximal activity concentration (AC_50_) or the area under the dose–response curve, Zamora et al. (2023) proposed a novel scoring approach that combines both potency and efficacy [104]. Two distinct scores were calculated: one reflecting a compound’s effect on a single cell line, and the other capturing the relative response difference between pNF and non-tumor Schwann cell lines. This approach successfully identified active compounds targeting known pathways such as RAF/MEK/ERK and PI3K-AKT, as well as unexpected targets including serotonin modulators, TOP2A and CHEK1 inhibitors, and heat shock protein inhibitors. These findings both reinforce prior drug discovery screening results and demonstrate the utility of integrated scoring systems for preclinical drug evaluation.

### 4.2. Cutaneous Neurofibromas

There have been significantly fewer drug screening efforts focused specifically on cNFs, likely due to only recent generation of patient-derived, immortalized cNF cell lines and the fact that cNFs are not life-threatening. Nonetheless, there has been a growing interest in the therapeutic investigation of cNFs in recent years.

In one study, NF1 patient-derived iPSCs were genetically modified using CRISPR and differentiated into Schwann cells to generate a cNF-like, *NF1*-homozygous cell line. This model was used to assess drug sensitivity in comparison to cNF tumor punches [47]. Differentiated *NF1^+/−^* and *NF1^−/−^* Schwann cell precursors were screened using a library of FDA-approved drugs to identify compounds that selectively inhibited the proliferation of *NF1^−/−^* cells. Econazole nitrate, a topical antifungal agent, was found to selectively kill *NF1^−/−^* cells, supporting its potential as a therapeutic candidate for cNFs.

More recently, patient-derived cNF organoids have been developed to create a more accurate *in vitro* model of cNFs for drug testing [61]. These organoids retain key molecular and cellular features of the original tumors. A small-scale proof-of-concept screen using kinase inhibitors in five cNF models identified several promising compounds as potential therapies for reducing cNF burden—copanlisib (PI3K inhibitor), onalespib (Hsp90 inhibitor), linsitinib (IGF-1R inhibitor), and digoxin (Na^+^/K^+^ ATPase inhibitor). This work is now being expanded to screen over 230 inhibitors across additional models [105].

## 5. MPNST Drug Screens

The most extensive NF1-related compound library screening efforts have focused on MPNSTs, utilizing a range of screening methodologies across multiple cellular models. In this section, we outline the diverse strategies employed in these HTS studies and the key discoveries that have emerged.

### 5.1. The First HTS Investigation in NF1-Derived MPNST Models

The first HTS drug discovery effort specifically targeting NF1-associated MPNSTs was performed in the *NF1*-deficient MPNST ST88-14 cell line and published in 2010. This study employed a chemical library screen using compounds from the National Cancer Institute (NCI) Diversity Set I [106]. The Diversity Set I, comprising 1990 small molecules, was designed to represent the broader NCI/NIH Developmental Therapeutics Program library of nearly 140,000 compounds [107]. From the screen, six compounds were selected for further evaluation, from which cucurbitacin-I (JSI-124) emerged as the lead compound. Cucurbitacin-I induced apoptotic cell death and suppressed JAK/STAT signaling in both MPNST cells and *NF1*-deficient astrocytes derived from *Nf1^fl/fl^* mice, which were tested to extend their findings to other *NF1*-deficient cell populations [106]. Interestingly, this study showed that NF1 regulates STAT3 phosphorylation via the mTOR pathway, as evidenced by use of inhibitors targeting PI3K (LY294002) and mTOR (rapamycin). *In vivo* administration of a commercial source of Cucurbitacin-I (JSI-124) into male athymic *nu/nu* mice injected subcutaneously with ST88-14 cells led to a significant reduction in *NF1*-deficient tumor growth compared to controls and supports STAT3 inhibition as a promising therapeutic strategy for NF1-associated tumors.

Following these investigations, the role of STAT3 in NF1 tumor development and progression was further explored, with STAT3 shown to contribute to neurofibroma initiation and growth by promoting macrophage recruitment and cytokine production within the tumor microenvironment [108]. The inhibition of STAT3 has also been shown to suppress tumor growth in both *NF1*-deficient pNFs and MPNSTs, while also reducing macrophage infiltration, inflammation, and cytokine expression, and enhancing apoptotic cell death *in vivo* [109,110,111].

### 5.2. HTS Provides NF1 MPNST Cell Model Profiles

Several large-scale HTS studies have been conducted using panels of commonly utilized cell lines, such as the NCI-60 and the Broad Institute’s Cancer Cell Line Encyclopedia (CCLE) [112]. However, disappointingly for NF1 research, these did not include frequently used MPNST cell lines, nor did they incorporate compounds specifically under investigation for treating MPNSTs. To address this gap, Guo et al. (2017) assembled a panel of the most widely studied MPNST cell lines, which included both NF1-associated and sporadic (non-NF1) patient-derived MPNST lines, enabling drug sensitivities specific to *NF1*-deficient cells to be explored [113]. These lines were screened against 130 highly relevant drugs, including most compounds under clinical investigation at that time. *NF1*-deficient MPNSTs exhibited distinct sensitivity to MEK and ERK inhibitors, BH3 mimetics, and FTIs, compared to the sporadic MPNST line. These findings provided strong preclinical support for MEK inhibition as a therapeutic strategy for NF1-associated MPNSTs and prompted further investigation of MEK inhibitors in combination with other targeted therapies [114,115].

Around the same time, Kolberg et al. conducted DSRT across multiple MPNST cell lines (three derived from NF1-related MPNSTs and two from sporadic MPNSTs), two primary cultures of human Schwann cells, and five normal bone marrow aspirates [116]. Their goal was to identify candidate therapeutics with minimal off-target toxicity. Of the 299 clinical and investigational drugs screened, Polo-like kinase 1 (PLK1) inhibitors and the antimetabolite gemcitabine emerged as promising therapeutic options. Subsequent analyses demonstrated that *PLK1* mRNA and protein levels were significantly elevated in MPNSTs compared to benign neurofibromas and normal Schwann cells, indicating a likely dependency on PLK1 in malignant cells. Analysis of patient data further revealed a correlation between high *PLK1* expression and poor prognosis.

This finding was reinforced by a separate study from [117], which explored PLK1 as a vulnerability in MPNSTs using a broader compound library (>2000 drugs) and an siRNA screen targeting most protein kinases in a single MPNST cell line (ST88-14) [117]. Both screening approaches independently identified PLK1 as a critical therapeutic target. Follow-up profiling of PLK1 inhibitors found them to be effective across six NF1-related MPNST lines (ST88-14, T265, ST88-3, 90.8, and S462TY) and one sporadic MPNST (STS-26T, now recognized as a melanoma line). One PLK1 inhibitor, BI6727, demonstrated disease stabilization and improved survival in an MPNST xenograft model. However, the authors cautioned that the clinical utility of PLK1 inhibitors as monotherapies may be constrained by their narrow therapeutic index.

While some MPNST screens focused on specific compound sets, others employed more diverse libraries. For example, Semenova et al. (2017) used the ICCB Known Bioactives library, consisting of 472 biologically active compounds developed in collaboration with the Harvard Institute of Chemistry and Cell Biology [118]. Initial screens in *Nf1^−/−^ E1A-p53* mouse embryonic fibroblasts (MEFs) and control cells (*Nf1wt E1A-p53*) identified only two compounds—cantharidin, a natural vesicant, and nifedipine, a calcium channel blocker—as warranting further investigation. These findings were validated in human MPNST cell lines, with S462TY cells showing marked sensitivity to nifedipine. *In vivo* studies using an S462TY xenograft mouse model confirmed that nifedipine significantly reduced tumor volume and weight, indicating that calcium channel inhibition may represent a viable therapeutic strategy for MPNSTs.

In addition to using established cell lines in screens, researchers have also created new panels of cell lines to better reflect the heterogeneity of NF1-associated tumors. Following extensive characterization, Oyama et al. (2020) performed a drug repurposing screen on their newly established cell lines, including four NF1 patient-derived MPNST cell lines, one sporadic MPNST line generated from surgical resections, and one line derived from a patient-derived xenograft (PDX) model [119]. The MPNST lines were then classified based on their response to the 164 anticancer drugs tested. A subset of drugs, including DNA-damaging agents, topoisomerase inhibitors, microtubule inhibitors, kinase inhibitors, histone deacetylase (HDAC) inhibitors, hormone signaling modulators, and proteasome inhibitors, was anti-proliferative in all lines, whereas others were active against subsets or non-functional. These results underscore the importance of heterogeneity of tumors for treatment response and indicate that follow-up studies with *in vivo* PDX models would be informative to determine the preclinical efficacy of the chosen compounds.

### 5.3. HTS Focused on Specific MPNST Characteristics

While unbiased compound library screening methods offer an objective approach to identifying novel drug targets, selectively targeting specific tumor characteristics can also uncover unique therapeutic opportunities. The epithelial–mesenchymal transition (EMT) is the process where epithelial cells lose their characteristic features and adopt a mesenchymal phenotype, increasing migratory and invasive abilities in tumor progression. A study by Harigai et al. (2018) focused on identifying targets within the EMT signaling cascade, which has previously been associated with neurofibromin loss [120,121]. EMT signaling can be induced in the human retinal pigment epithelial cell line, ARPE-19, treated with tumor necrosis factor-α (TNF-α) and transforming growth factor-β2 (TGF-β2), resulting in cell aggregation, or focus formation [122]. This cell system was screened with a library of over 1500 FDA-approved compounds to identify inhibitors of EMT, identifying eight candidates [121]. The group went on to show that tranilast, an anti-allergenic drug, suppressed EMT characteristics in an *NF1*-deficient MPNST cell line (sNF96.2), inhibiting cell growth *in vitro* and in a xenograft mouse model. *In vitro* analyses also showed that tranilast suppressed expression of mesenchymal markers and angiogenesis-related genes, suggesting its potential to inhibit NF1-associated tumor growth through EMT and angiogenesis suppression [121].

### 5.4. Combinatorial Inhibition as a Therapeutic Option in MPNSTs

The most efficient approach to advancing a therapy through the clinic for a condition like NF1 is to repurpose existing FDA-approved drugs or compounds already in clinical development. In one study, 59 drugs were evaluated—both individually and in combination—across multiple MPNST cell line models to identify therapies with superior efficacy compared to the current standard of care [123]. This initial screen highlighted several promising therapeutic classes, including clinically used cytotoxic agents, RAS pathway inhibitors, HDAC inhibitors, and microtubule-targeting agents.

Subsequent combinatorial analyses involved testing 72 two-drug combinations across four cell lines at 25 different concentrations. These experiments aimed to identify combinations with the highest therapeutic effect (fraction affected) and the lowest combination index scores. Among the combinations tested, dual inhibition of MEK and mTORC1/2 emerged as the most effective strategy for suppressing RAS signaling in MPNST cell lines, suggesting this combination may offer clinical benefit for NF1 patients with MPNSTs [123].

In a separate combinatorial drug screen, compounds from the NCATS MIPE library were tested across three MPNST cell lines *in vitro*, followed by validation of top-performing combinations in patient-derived xenograft (PDX) models of primary MPNSTs [124]. Of the 21 synergistic combinations identified, four demonstrated strong efficacies across a broad panel of MPNST lines. Combination of the WEE1 inhibitor MK-1775 and the chemotherapy agent doxorubicin significantly reduced tumor growth in models of both NF1-associated and *TP53*-mutant MPNSTs, as well as in a sporadic MPNST model. These findings suggest a combinatorial therapeutic strategy not only for *NF1*-mutant MPNSTs, but also for individuals with sporadic tumors.

In a third combinatorial drug screen, Ortega-Bertran et al. (2024) evaluated a subset of inhibitors from the NCATS MIPE 4.0 library against pathways commonly dysregulated in MPNSTs [125]. Screening targets included MEK, CDK, and the bromodomain and extra-terminal domain (BET) family. Loss of *SUZ12* or *EED* in MPNSTs causes an epigenetic change, resulting in elevated global levels of H3K27 acetylation. A consequent increase in recruitment of BET family of epigenetic reader proteins, such as BRD4, upregulates transcription of c-*MYC*, *TP53*, or members of the Ras pathway, leading to hyperproliferation [126]. Inhibitors targeting these pathways were tested as single agents, dual combinations, and triple combinations in three NF1-related MPNST cell lines (S462, NF1-08, and NF1-09) to identify potential therapeutic strategies. From this screen, 22 lead compounds were prioritized and evaluated across a panel of NF1-related and sporadic MPNST cell lines, yielding 147 treatment conditions. Dose response analysis identified eight compounds in 21 synergistic combinations for further validation: three MEK inhibitors (ARRY-162, selumetinib, trametinib), three CDK inhibitors (palbociclib, R-547, flavopiridol), two BET inhibitors (JQ1, I-BET151), and the FDA-approved CDK inhibitor (ribociclib), which was not included in the MIPE 4.0 library. The most effective dual combination (ARRY-162 + I-BET151) and triple combination (ARRY-162 + I-BET151 + ribociclib) were advanced into NF1-associated and sporadic MPNST patient-derived orthotopic xenograft (PDOX) models. Dual MEK/BET inhibition reduced tumor burden by ~75%, while adding CDK inhibition achieved an additional 50% reduction compared with MEK/BET therapy alone. In the sporadic MPNST PDOX model, the same regimens reduced tumor volume by 65% (dual) and 85% (triple). Together, these results provide strong preclinical evidence that combined MEK, BET, and CDK inhibition represents a promising therapeutic strategy for both NF1-associated and sporadic MPNSTs.

### 5.5. Pharmacoproteomics Studies to Unveil Novel Targets in MPSNTs

While HTS drug discovery is a powerful stand-alone method for identifying potential therapeutic options in specific tumor types, integration with proteomics enhances its utility. By accounting for protein expression levels, this combined approach can help overcome the frequent challenge of poor *in vitro* to *in vivo* translatability observed in many screening studies [127]. In one such study, researchers conducted an in-depth proteomic analysis of 23 MPNST samples from NF1 patients, alongside an HTS of 214 drugs across six distinct MPNST cell lines. Of the 4650 proteins detected, 68 were significantly upregulated, leading to the identification of 70 pathways, two of which were deemed druggable. Additionally, 24 drugs effectively reduced MPNST cell viability, and 14 targets were identified based on their mechanisms of action. Integrating these datasets led to the identification of crizotinib and foretinib—two MET pathway inhibitors—as promising therapeutic candidates for treating MPNSTs, irrespective of *NF1* status.

### 5.6. Rapid In Vitro to In Vivo Small Molecule Screening in NF1-Relevant Models

A recent high-throughput drug screen was conducted to improve preclinical screening strategies using isogenic CRISPR gene-edited immortalized human Schwann cells [49]. A library of 11,085 small molecules—comprising FDA-approved drugs, clinically advanced compounds, and drug-like molecules—was screened in an *NF1*-homozygous Schwann cell line harboring an indel in exon 10. Compounds active in this primary screen were further evaluated in a matched pair of CRISPR-derived *NF1* heterozygous and homozygous Schwann cell lines. This secondary screen identified 27 compounds that were selectively lethal to *NF1*-deficient cells. These hits were classified into several drug categories, with four mechanistic classes prioritized for further study: agents that modulate intracellular Ca^2+^ levels, induce ER stress/unfolded protein response, affect NF-κB or p53 pathways, promote cell cycle arrest, or disrupt microtubule stability via PLK-1 inhibition. Representative compounds from each class were advanced to *in vivo* testing using MPNST mouse models, either as monotherapies or in combination with MEK inhibition. Of note, the microtubule stability/PLK-1 inhibitor rigosertib and the intracellular Ca^2+^ enhancer digoxin showed efficacy in reducing MPNST growth—both as single agents and in combination with selumetinib. These findings highlight potential therapeutic strategies for MPNST as well as underscoring the utility of this rapid HTS approach for evaluating novel compounds and drug combinations in future studies.

### 5.7. HTS in an In Vivo MPNST Model Organism

An *nf1/tp53*-deficient MPNST zebrafish model has been used to screen a series of candidate agents for their ability to induce apoptosis [69]. Among the compounds tested, DNA topoisomerase I inhibitors and mTOR kinase inhibitors emerged as the most effective single agents, selectively eliminating MPNST cells without causing prohibitive toxicity in the zebrafish. Notably, irinotecan triggered apoptosis by activating a DNA damage response and acted synergistically with AZD2014 to induce hypo-phosphorylation of 4E-BP1, leading to protein synthesis arrest and tumor cell death. These findings not only identified DNA topoisomerase I and mTOR as synergistic therapeutic targets in *NF1*-deficient tumors but also demonstrated the utility of the zebrafish model for drug discovery in MPNST research.

## 6. Functional Genomic and Small Molecule Screens to Identify New Therapeutic Targets for NF1-Associated Tumors

### 6.1. In Vivo Transposon Screen to Validate MPNST-Associated Pathways

While biallelic loss of *NF1* is a hallmark of many MPNSTs, other tumor suppressor genes are inactivated during MPNST transformation. Understanding the multiple pathways directly affected by tumor suppressor loss will be critical for identifying new effective therapeutic strategies. One approach that has been taken to identify genetic drivers of MPNST development is using the Sleeping Beauty (SB) transposon system, which can disrupt random genes via insertional mutagenesis in mice. The *Cnp-Cre* transgene, combined with a conditional Sleeping Beauty (SB) transposon mutagenesis system, was used to specifically target Schwann cells and their precursors. This strategy has enabled the identification of candidate oncogenes and tumor suppressor genes by assessing whether gene disruption promoted peripheral nerve sheath tumor formation in mice [128].

In a follow-up HTS, 103 candidate genes were further assessed for their roles in anchorage-independent growth and cellular migration. A subset of these genes was subsequently tested for tumorigenic potential *in vivo* [60]. The study found that disruption of nearly 60% of candidate tumor suppressor genes induced transformation of immortalized Schwann cells, while knockdown of approximately 30% of candidate oncogenes impaired the growth of MPNST cell lines. Notably, independent loss of four genes—including *NF1*—resulted in upregulation of both the Wnt and Hippo signaling pathways. These findings suggest that therapeutic strategies targeting Wnt and Hippo signaling may be particularly effective in treating MPNSTs, especially in the context of *NF1* loss.

### 6.2. Synthetic Lethal Screens to Identify Novel Therapeutic Targets in NF1-Associated Tumors

The first exploration of harnessing synthetic lethality through high-throughput screening (HTS) in the context of NF1 was reported in 2011 by Wood and colleagues, who used *Saccharomyces cerevisiae* (budding yeast) lacking *ira2*, the yeast ortholog of the human NF1 gene. In this work, 6000 structurally diverse compounds, selected to represent a larger library of over 340,000 drug-like molecules maintained by the University of Cincinnati Drug Discovery Center, were screened against wild-type and *ira2Δ* yeast. The screen identified several lead tool compounds with selective activity against *ira2Δ* yeast [54], with one of these compounds, UC1, selected for further investigation. A high-copy suppressor screen was employed in *ira2Δ* yeast to identify Nab3 as a plausible molecular target of UC1. Follow-up *in vitro* experiments using *NF1* wild-type (*NF1^+/+^*) and KO (*NF1^−/−^*) MPNST cell lines confirmed that UC1 sensitivity observed in yeast translated to human cells. Wood et al. also identified Ctk1 as another potential target in *ira2Δ* yeast, as combinatorial loss of *ira2* and *ctk1* was synthetically lethal. Both Nab3 and Ctk1 are associated with the RNA Pol II C-terminal domain, suggesting that UC1 treatment could be impacting RNA Pol II function. In support of this, KO of *ctk1* further sensitized cells to UC1. Based on these observations, CDK9, the yeast ortholog of Ctk1, has been proposed as a therapeutic target in *NF1*-deficient MPNSTs.

Supporting this hypothesis, CDK inhibitors were identified through large-scale drug screening and RNA-seq analysis as candidates for overcoming MEK inhibitor resistance in pNFs [129]. Combined treatment with dinaciclib (a CDK inhibitor) and TAK-733 (a MEK inhibitor) significantly reduced viability of pNF-derived immortalized Schwann cell lines (ipNF95.11b C and ipNF05.5). Moreover, in a PDX mouse model of pNF, this combination induced tumor regression, supporting the clinical potential of dual MEK/CDK inhibition.

Following the initial yeast screen, two additional tool compounds have been studied in greater depth. The compound Y100 and its analog, Y100B, were shown to reduce cell viability in at least two *NF1*-deficient glioblastoma cell lines [130]. Mechanistic studies revealed that Y100 disrupted proteostasis and metabolic homeostasis while inducing mitochondrial superoxide production. These findings indicated that *NF1*-deficient cells are particularly vulnerable to oxidative stress, endoplasmic reticulum (ER) stress modulators, and mitochondrial disruptors.

In a separate study, the tool compound Y102 was found to reduce viability across multiple *NF1*-deficient cell lines, including glioblastoma, pNF, and MPNST models [131]. Y102 treatment perturbed autophagy, mitophagy, and lysosome positioning specifically in *NF1*-deficient cells. Using a proteomics strategy, the molecular target of Y102 was identified as BORCS6, a subunit of the BORC complex. The BORC complex regulates lysosomal trafficking following autophagosome fusion, and this study was the first to implicate BORC as a potential vulnerability in *NF1*-deficient and RAS-dysregulated cells.

Yeast is not the only model organism to have been employed in synthetic lethal HTS. Stevens et al. (2025) utilized a genetic HTS using an RNA interference (RNAi) library comparing effects between *dNf1* wild-type and CRISPR *dNf1* KO *Drosophila* Schneider 2 cell lines [50]. This screen identified five candidate pathways amenable to being targeted with existing drugs. Among these, compounds targeting the autophagy pathway (chloroquine (CQ) and bafilomycin A1) displayed the greatest selective lethality in *dNf1*-deficient *Drosophila* cells. Furthermore, CQ treatment also significantly reduced tumor growth in a xenograft *NF1* mouse model, outperforming selumetinib. These findings highlight the therapeutic promise of autophagy inhibition, and CQ in particular, in treating *NF1*-deficient tumors.

## 7. NF1-Associated Cancers

In addition to neurofibromas, individuals with NF1 are predisposed to other tumor types, particularly astrocytomas and gliomas. This is likely due to the high expression of neurofibromin in neuronal cells, where it plays a critical role in regulating RAS activity. Astrocytomas, which are sometimes precursors to glioblastomas (GBMs), develop in approximately 15–20% of NF1 patients and are associated with a poor 5-year survival rate of ~27% [132]. While GBMs occur less frequently (in ~7% of NF1 patients), they have an even more dismal 5-year survival rate of just 5% [133]. Despite their clinical significance, the molecular mechanisms driving these tumor types in the context of NF1 remain poorly understood. Here, we briefly highlight large-scale drug screening strategies aimed at identifying therapeutic targets for astrocytomas and GBMs in NF1.

### 7.1. Astrocytomas

While natural product libraries do exist, they are less frequently explored for novel treatment options compared to drug repurposing strategies, which offer a faster path to FDA approval. In this HTS, however, a unique collection of 68,427 pre-fractionated and partially purified natural product extracts was screened against *Trp53*/*Nf1* mutant astrocytoma cells derived from the *Nf1^−/+^*; *Trp53^−/+^* cis mouse model [134]. A total of 95 unique extracts demonstrated activity in reducing proliferation of *Nf1*-deficient cells in their partially purified form, while only three extracts retained this activity in their crude form. One of these active extracts, obtained from a Vietnamese collection of *Millettia ichthyotona* (commonly known as Thàn mát), yielded two active compounds upon further purification: deguelin and dehydrodeguelin. Although deguelins have previously been studied as potential cancer therapeutics, their effects in the context of NF1 had not been previously investigated [135]. Both compounds appeared to selectively inhibit proliferation in mouse *Nf1*-deficient astrocytoma cell lines without affecting normal primary astrocytes or astrocytes with heterozygous *Nf1* loss. These findings could suggest that deguelin, dehydrodeguelin, or related analogues may warrant further investigation as potential targeted therapies for NF1-associated astrocytomas.

### 7.2. Glioblastomas

One major limitation of screening methods is that *in vitro* results have often failed to translate into *in vivo* efficacy. To address this disconnect, researchers are taking advantage of 3D culture systems and spheroid-based screens. In one recent HTS, patient-derived glioblastoma (GBM) spheroid models were used to identify drivers of GBM progression and therapeutic targets relevant to both *NF1*-mutant and *NF1*-WT tumors [136]. Among nine spheroid lines analyzed, four were driven by *NF1* loss, with additional key alterations including *PTEN* and *CDKN2A*. Subsequently, one *NF1*-deficient and one *NF1*-WT line were screened against 1912 compounds, both as single agents and in combination. While both lines responded to Hsp90 and proteasome inhibitors, the *NF1*-deficient line showed selective sensitivity to MEK inhibition. Combination screening revealed a synergy between PI3K inhibitors and either MEK or proteasome inhibitors; however, *in vivo* testing showed that MEK inhibition alone produced superior efficacy compared to combination therapy.

A genome-wide CRISPR interference (CRISPRi) screen to identify potential targets that synergize with MEK inhibition in *NF1*-mutant glioblastoma (GBM) cells has recently been reported [137]. Initial screening of untreated cells at two timepoints revealed genes essential for tumor growth, with strong enrichment of cell cycle regulators. A subsequent screen compared selumetinib- and vehicle-treated cells after 10 days. In both mouse SB28 and human GBM43 cell lines, *BRAF* and *SHOC2* emerged as top hits with increased sgRNA abundance when comparing selumetinib-treated cells to vehicle-treated cells of the same cell type. While *BRAF*’s role in gliomagenesis is well established, this study is the first to implicate *SHOC2*—a key modulator of RAF signaling—as a potential therapeutic target in *NF1*-mutant GBM. *In vivo* validation demonstrated that *SHOC2* suppression enhanced the efficacy of selumetinib in intracranial GBM models, resulting in more durable responses than either treatment alone. Although two clinical trials investigating MEK inhibition in GBM are currently underway or recently completed [138,139], the combination strategies explored in those trials differ from those identified in these studies.

Given the central role of MEK signaling in the pathogenesis of HGGs and the clinical success of MEK inhibitors in treating pNFs, many studies have focused on identifying effective combination therapies involving MEK inhibition. In one small-scale *in vitro* screen, 21 clinically relevant cancer drugs were tested alone or in combination with MEK or PI3K inhibitors in both 2D and 3D cultures of *NF1*-deficient HGG cells [140]. The cell lines used were derived from either an *NF1* patient with a malignant astrocytoma or *Nf1*-deficient mice that spontaneously developed HGGs [140,141]. A total of six compounds targeting HDAC (Vorinostat), BRD4 (JQ1), CHK1 (LY2606368), BMI1 (PTC596), CDKs (Dinaciclib), and the proteasome (Bortezomib) induced cell death in both models. Among them, JQ1, LY2606368, and PTC596 demonstrated synergy with MEK inhibition, while Vorinostat, LY2606368, and PTC596 showed synergy with PI3K inhibition.

A recent 3D GBM model employed CRISPR-Cas9 gene editing to introduce different GBM driver mutations into WT human iPSCs, which were then used to generate cerebral organoids for comprehensive multi-omics analyses [48]. Three GBM organoid models were developed, including one with *NF1* loss. Multi-layered profiling—including single-cell transcriptomics, DNA methylomics, metabolomics, lipidomics, proteomics, and phosphoproteomics—revealed that reprogramming of glycerolipid metabolism was a consistent hallmark across all models. Notably, *NF1* loss specifically drove a mesenchymal signature. Guided by these omics data, a targeted screen was performed using 327 compounds including FDA-approved, blood–brain barrier-permeable drugs. Lomitapide, an inhibitor of microsomal triglyceride transfer protein, emerged from the screen as a potent inhibitor of growth in all three organoid models. Furthermore, lomitapide reduced tumor cell proliferation and significantly prolonged survival in at least one *in vivo* GBM model, suggesting that targeting lipid metabolism may represent a promising therapeutic strategy for both *NF1*-mutant and *NF1* wild-type GBMs.

Consideration of the genetic heterogeneity of NF1 tumors is critical for developing new therapeutic approaches. To take this aspect into account, several screens have been conducted with *NF1*-deficient cell lines with co-mutations such as *TP53*, *PTEN*, and *CDKN2A/B* [48,66,136,137]. One large HTS of 10,000 small molecules focused on the co-mutation of *NF1* and *ATRX*, a combination associated with a more aggressive GBM phenotype [142]. This screen utilized an established *NF1*-mutant cell line (U251), an isogenic U251 *ATRX*-KO variant, and a patient-derived *NF1*/*ATRX*-mutant line (JHH-NF1-GBM1). From the initial hits, 105 compounds demonstrated selective efficacy in the *NF1/ATRX* co-mutant context. Further analysis of biological activity and target-binding interactions narrowed this list to eight candidate compounds for validation. One lead tool compound, K784-6195, was identified, and mechanistic studies revealed that it disrupts redox homeostasis. While K784-6195 itself is not clinically translatable, these findings indicate that existing redox modulators and oxidative stress inducers may represent promising therapeutic strategies for treating *NF1/ATRX*-mutant GBMs, a particularly aggressive HGG subtype.

## 8. Translational Efforts and Clinical Trials for NF1-Associated Tumors

To date, few HTS efforts have resulted in direct clinical translation; however, several targets identified through these screens have advanced to clinical trials in NF1-related tumor types, where therapeutic hits are being pursued using the original screening compound, newer inhibitors, or FDA-approved drugs (Table 1). Up-to-date information on the latest clinical trials for NF1 can be found at https://clinicaltrials.gov/.

For example, while mTOR inhibition was identified as a potential therapeutic vulnerability in MPNSTs through HTS [106] as well as multiple non-HTS studies [143,144,145], testing of mTOR inhibitors such as everolimus and sirolimus (also known as rapamycin) has shown limited efficacy as single agents in pNFs and cNFs [99,146,147,148]. Preclinical *in vivo* analysis of mTOR inhibition demonstrated that prolonged treatment of MPNSTs with everolimus as a single agent resulted in the development of resistance and mTOR pathway reactivation [149]. Additionally, a recent Phase II clinical trial for pNFs evaluated cabozantinib, a tyrosine kinase inhibitor that, while not a direct mTOR inhibitor, may exert indirect effects through downstream signaling blockade [150]. It is now under investigation as a combination therapy with the MEK inhibitor Selumetinib to improve pNF reduction in NF1 young and mature adults (NCT06502171). Direct mTOR inhibitors are also currently under investigation as single agents in NF1-associated LGGs [100] and in combination therapies [151,152,153,154]. For instance, bevacizumab, an anti-angiogenic agent, has been studied in combination with everolimus for MPNSTs [151].

Ongoing and recently completed clinical trials are aimed at evaluating additional MEK inhibitors, such as trametinib and binimetinib, in both pediatric and adult populations for various NF1-associated tumors [150,155,156,157,158] (NCT03741101, NCT03231306). Multiple clinical trials have also necessarily explored combination therapy with MEK inhibition, as monotherapies of other inhibitors have been shown to rarely achieve durable responses. These include testing MEK inhibitors in combination with mTOR inhibitors in MPNSTs [101], BRAF inhibition in pediatric HGGs [159], BRAF and autophagy inhibitors in LGGs or HGGs (NCT04201457), HDAC inhibition in MPNSTs (NCT06693284), MDM2 inhibition in patients with NF1-associated MPNSTs (NCT06735820), and bromodomain (BET) inhibition in MPNSTs (NCT05253131). More traditional chemotherapy has also been tested with respect to NF1-related and sporadic MPNST treatment in combination with DNA-damaging agents [160].

Outside of combination therapy, there are a few clinical trials that have previously or are actively exploring single-target therapies outside of direct MEK inhibition. These include CDK4/6 inhibition in atypical neurofibromas (NCT04750928) and tyrosine kinase inhibition in MPNSTs [161]. It is important to note that, while not directly stated in the clinical trial designs, several of these targeting strategies were identified through an HTS even if not for the specific symptom being investigated; these include proteasomal inhibition [119,136], mTOR inhibition [69], DNA damaging agents, multi-kinase inhibitors, and HDAC inhibition [119], BRAF inhibition [137], and combinatorial MEK and mTOR inhibition [123].

Several challenges hinder clinical translation, including tumor heterogeneity and the slow growth of pNFs, which complicate the definition of clinical trial endpoints. Limited access to biopsy tissue further restricts molecular characterization, underscoring the need for improved biomarkers and advanced imaging techniques to monitor treatment response. NF1 tumors vary widely in growth rate, anatomical location, and malignant potential, necessitating personalized therapeutic strategies. Currently, predictive biomarkers for treatment response are lacking, but assessment of *NF1* mutation status, pathway activation profiles, and liquid biopsy approaches like circulating tumor DNA (ctDNA) hold promise for guiding therapy. Drug delivery also remains a significant obstacle for NF1-associated tumors, as many pNFs and MPNSTs reside in deep or surgically inaccessible areas. Innovative delivery methods, including nanoparticles and intralesional injections, are being explored to enhance therapeutic efficacy and precision.

**Table 1 curroncol-32-00649-t001:** Status of drugs identified through high-throughput screening in NF1.

Drug Name	Target	Cell Type Efficacy	Status	Monotherapy/ Combinatorial	Screen
10-hydroxycamptothecin	TOP1	GBMs	Further study needed	-	[136]
5,15-DPP	STAT3	MPNSTs	Further study needed	-	[109]
A-443654	AKT1	pNFs	Further study needed	-	[104]
Actinomycin D	TOP2A/B, TOP1	MPNSTs	Further study needed	-	[119,127]
Afatinib	EGFR, HER2, HER4	MPNSTs	Further study needed	-	[119]
Alvespimycin HCl	HSP90AB1	pNFs	Further study needed	-	[104]
Aminopterin	PDF	GBMs	Further study needed	-	[136]
APX3330	APE1/Ref-1	MPNSTs	Further study needed	-	[111]
Arsenic (III) Oxide	IKBKB, TXNRD1, JUN, CCND1, MAPK3, MAPK1	MPNSTs	Further study needed	-	[119]
AT-7867	AKT1	pNFs	Further study needed	-	[104]
AT-9283	AURKA/AURKB	MPNSTs	Further study needed	-	[124]
AZ628	RAF	MPNSTs	Further study needed	-	[113]
AZD-8330	MAP3K1	GBMs	Further study needed	-	[136]
AZD2014	mTOR	MPNSTs	Further study needed	-	[69]
AZD8055	mTOR	MPNSTs	Further study needed	-	[113,136]
Bafilomycin A1	V-ATPase	pNFs	Further study needed	-	[50]
Bardoxolone methyl	NFKBIA	GBMs	Further study needed	-	[136]
Belinostat	HDAC	MPNSTs	Further study needed	-	[119,127]
Bergapten	NLRP3, Pyroptosis	MPNSTs	Further study needed	-	[119]
BI-847325	MEK/AURKC	MPNSTs	Further study needed	-	[113]
BI2536	PLK1	Melanoma ^#^, MPNSTs	Phase II Study (NCT00526149- Completed)	Monotherapy	[116]
BIIB021	HSP90AA1	GBMs	Further study needed	-	[136]
Binimetinib/ARRY-162	MEK	pNFs, MPNSTs	Phase II Study (NCT03231306- Completed)	Monotherapy	[125]
BKM120	PI3K	MPNSTs	Further study needed	-	[123]
BMS-186511	FTI	MPNSTs	Other FTIs tested; Phase II Study (NCT00021541- Completed), Phase II Study (NCT00076102 Completed)	Monotherapy	[78]
Bortezomib	Proteosome	pNFs	Further study needed	-	[42,119,127,140]
Cabozantinib	VEGFR2	pNFs, MPNSTs	Phase II Study (NCT02101736- Completed); Phase I Study (NCT06502171-Not Yet Recruiting)	Monotherapy & Combinatorial	[124]
Camptothecin	TOP1	GBMs, MPNSTs	Further study needed	-	[127,136]
Cantharidin	PP2A	MPNSTs	Further study needed	-	[118]
Carboplatin	Alkylating Agent	pNFs, LGGs, MPNSTs	Phase I Study (NCT00352495- Completed); Phase III Study (NCT03871257 Active, not recruiting); Phase III Study (NCT04166409-Recruiting)	Combinatorial	[119]
Carfilzomib	Proteosome	MPNSTs	Further study needed	-	[124]
Cephalomannine	HIF1A, APEX1, BCL2L1, MAPK14, SYK, TNF, ADAM17	MPNSTs	Further study needed	-	[119]
Ceritinib	ALK	MPNSTs	Further study needed	-	[127]
Chloroquine/ Hydroxychloroquine	Lysosomal Alkalinizing Agent	pNFs, LGGs	Phase I/II Study (NCT04201457-Active, not recruiting)	Combinatorial	[50]
CI-1040	MEK1/2	GBMs	Further study needed	-	[136]
Cladribine	DCK	GBMs, MPNSTs	Further study needed	-	[119,136]
Clofarabine	RNR	MPNSTs	Further study needed	-	[119]
Clomifene Citrate	ESR1/2	MPNSTs	Further study needed	-	[119]
Clomipramine HCl	Serotonin	pNFs	Further study needed	-	[104]
Cobimetinib	BRAF	MPNSTs	Further study needed	-	[123]
Copanlisib	PI3K	cNFs	Further study needed	-	[61]
Crizotinib	ALK/ROS1	MPNSTs	Further study needed	-	[113,119,127]
Cucurbitacin-I	JAK/STAT3	pNFs, MPNSTs	Further study needed	-	[106,108,109,111]
Dasatinib	BCR-ABL, RTKs	MPNSTs	Further study needed	-	[119]
Daunorubicin	TOP2A/B	pNFs, MPNSTs	Phase II Study (NCT00304083- Completed)	Combinatorial	[119,127]
Defactinib	RHO, FAK, Pyk2	MPNSTs	Further study needed	-	[60]
Deguelin	AKT	Astrocytomas	Further study needed	-	[134]
Dehydrodeguelin	CI	Astrocytomas	Further study needed	-	[134]
Deltarasin	PDEδ	MPNSTs	Further study needed	-	[113]
Digoxin	Na^+^/K^+^ ATPase	cNFs, MPNSTs	Further study needed	-	[49,61]
Dinaciclib	CDK1/2/5/9	pNFs	Further study needed	-	[129,140]
Doxorubicin	TOP2A/B, TOP1	pNFs, MPNSTs	Phase II Study (NCT00304083- Completed)	Combinatorial	[104,119,124,127]
Duloxetine HCl	Serotonin	pNFs	Further study needed	-	[104]
Econazole Nitrate	Lanosterol 14-alpha Demethylase	cNFs	Further study needed	-	[47]
Elesclomol	Copper Chelator	MPNSTs	Further study needed	-	[124]
Entrectinib	ROS1/TRK1	MPNSTs	Further study needed	-	[127]
Enzalutamide	AR	MPNSTs	Further study needed	-	[119]
Epirubicin	TOP2A/B	pNFs	Further study needed	-	[104,119,127]
Erastin	VDAC2/VDAC3	MPNSTs	Further study needed	-	[113]
FLLL31	JAK2/STAT3	MPNSTs	Further study needed	-	[109]
FLLL32	JAK2/STAT3	pNFs	Further study needed	-	[108,110]
Fluvastatin	HMGCR, HDAC	pNFs	Further study needed	-	[104]
Foretinib	MET/VEGFR	MPNSTs	Further study needed	-	[127]
Fostamatinib	SYK	MPNSTs	Further study needed	-	[127]
GDC-0941	PIK3CG	GBMs	Further study needed	-	[136]
GDC-0973	MAP2K1	GBMs	Further study needed	-	[136]
Geldanamycin	HSP90AB1	pNFs	Further study needed	-	[104]
Gemcitabine	DNA Synthesis	Melanoma ^#^, MPNSTs	Phase IB/II Study (NCT01418001- Terminated); Phase II Study (NCT01532687-Completed)	Combinatorial	[116]
GGTI-2Z	GGTI	MPNSTs	Further study needed	-	[86]
GNE-490	PI3K	pNFs	Further study needed	-	[104]
GSK-2126458	PI3K/mTOR	GBMs	Further study needed	-	[136]
GSK-2636771	PI3K	pNFs	Further study needed	-	[104]
GSK-461364	PLK1	Melanoma ^#^, MPNSTs	Further study needed	-	[116]
Homoharringtonine	RPL2/3	MPNSTs	Further study needed	-	[119,127]
I-BET151	BRD2/3/4	MPNSTs	Further study needed	-	[125]
Idarubicin	TOP2A/B	pNFs	Further study needed	-	[104,119,127]
IKK-2 Inhibitor VII	IKK	pNFs	Further study needed	-	[104]
Imatinib	TRK	pNFs, MPNSTs	Phase I/II Study (NCT01140360- Completed); Phase II Study (NCT02177825- Terminated); Phase II Study (NCT01673009-Completed); Phase II Study (NCT03688568-Withdrawn); Phase II/III Study (NCT00427583- Terminated)	Monotherapy & Combinatorial	[108]
INK128	mTOR	MPNSTs	Further study needed	-	[69,123]
Irinotecan	TOP1	MPNSTs	Further study needed	-	[69]
Isradipine	Calcium Channels	pNFs	Further study needed	-	[104]
JNK inhibitor IX	JNK	MPNSTs	Further study needed	-	[113]
JQ1	BRD4	MPNSTs	Further study needed	-	[113,140]
Ketorolac	COX	pNFs	Further study needed	-	[104]
Lamotrigine	Sodium Channels	pNFs, MPNSTs	Phase II Study (NCT03504501- Terminated); Phase II/III Study (NCT02256124- Terminated)	Monotherapy	[119]
LDN-193189	ALK	pNFs	Further study needed	-	[103]
Linagliptin	DPP4	MPNSTs	Further study needed	-	[119]
Linsitinib	IGF1R	cNFs	Further study needed	-	[61]
Lomitapide	Cytochrome P450 3A4	GBMs	Further study needed	-	[48]
Lovastatin	HMG-CoA	MPNSTs	Tested for NF1-related cognitive deficits (NCT00352599, NCT00853580), autism spectrum disorder (NCT03826940), and reading disabilities (NCT02964884)	Monotherapy	[84,85,86]
LY2606368	CHEK1	HGGs	Further study needed	-	[140]
LY3009120	RAF	MPNSTs	Further study needed	-	[113]
Marizomib	Proteasome	GBMs	Further study needed	-	[136]
Mirdametinib/PD0325901	MEK	pNFs, GBMs, MPNSTs	FDA-approved for pNFs	Monotherapy	[113,136]
Mitomycin C	Alkylating Agent	MPNSTs	Further study needed	-	[119,127]
Mitoxantrone	TOP2A/B	pNFs	Further study needed	-	[104,127]
MK-1775	WEE1	MPNSTs	Further study needed	-	[124]
MLN8237	AURKA	MPNSTs	Further study needed	-	[117]
Mycophenolic Acid	IMPDH	pNFs	Further study needed	-	[104]
Napabucasin	STAT3	MPNSTs	Further study needed	-	[111]
Neratinib	EGFR	MPNSTs	Further study needed	-	[124]
Niclosamide	STAT	MPNSTs	Further study needed	-	[113]
Nifedipine	Calcium Channels	MPNSTs	Further study needed		[118]
NVP-BGT226	PI3K	MPNSTs	Further study needed	-	[124]
Onalespib	HSP90	cNFs	Further study needed	-	[61]
Osimertinib	EGFR	MPNSTs	Further study needed	-	[127]
Panobinostat	HDAC	MPNSTs	Further study needed	-	[124]
PD-318088	MEK	GBMs	Further study needed	-	[136]
PF-04217903	c-Met	MPNSTs	Further study needed	-	[109]
PF-3758309	PAK1/2/3/4/5/6	MPNSTs	Further study needed	-	[113]
PF04691502	PI3K/mTOR	MPNSTs	Further study needed	-	[113]
Piboserod HCl	Serotonin	pNFs	Further study needed	-	[104]
Picropodophyllin	IGF1R	pNFs	Further study needed	-	[103]
Ponatinib	FGFR	MPNSTs	Further study needed	-	[119,124,127]
PTC596	BMI1	HGGs	Further study needed	-	[140]
R-1487	p38α	pNFs	Further study needed	-	[104]
Rapamycin/Sirolimus	mTOR	pNFs, cNFs, LGGs, MPNSTs	Phase I Study (NCT01031901- Completed); Phase I Study; (NCT00901849-Completed); Phase II Study (NCT00634270 -Completed); Phase II Study (NCT03433183-Completed)	Monotherapy & Combinatorial	[69,109]
Retaspimycin	HSP90AB1	pNFs	Further study needed	-	[104]
Ribociclib	CDK4/6	MPNSTs	Further study needed	-	[125]
Rigosertib	PLK1, PI3K	Melanoma ^#^, MPNSTs	Further study needed	-	[49,116]
RMC-7977	RAS	Gliomas, MPNSTs	Further study needed	-	[87]
Romidepsin	HDAC	MPNSTs	Further study needed	-	[119,127]
Ruxolitinib	STAT3	MPNSTs	Further study needed	-	[111]
SCH-900776	CHK1	pNFs	Further study needed	-	[104]
Selumetinib	MEK	pNFs, MPNSTs	FDA-approved for pNFs	Monotherapy	[113]
SH-4-54	STAT3/5	MPNSTs	Further study needed	-	[113]
Sibutramine HCl	Serotonin	pNFs	Further study needed	-	[104]
SN-38	TOP1	GBMs	Further study needed	-	[136]
Sorafenib	RAF	pNFs, MPNSTs	Phase I Study (NCT00727233- Completed)	Monotherapy	[113,119]
SU11274	c-Met	MPNSTs	Further study needed	-	[109]
Sunitinib	RTKs	pNFs, MPNSTs	Phase II Study (NCT01402817- Terminated)	Monotherapy	[68,119]
TAK-285	HER2/EGFR	pNFs	Further study needed	-	[104]
TAK-632	RAF/VEGFR	MPNSTs	Further study needed	-	[113]
TAK-733	MEK	pNFs, GBMs	Further study needed	-	[129,136]
Teniposide	TOP2A/B	MPNSTs	Further study needed	-	[127]
Thapsigargin	SERCA	MPNSTs	Further study needed	-	[124]
Tipifarnib	FTI	MPNSTs	Phase II Study (NCT00021541- Completed)	Monotherapy	[113]
Tivozanib	VEGFR	pNFs	Further study needed	-	[104]
Topotecan	TOP1	MPNSTs	Further study needed	-	[127,136]
Torin-2	mTORC	MPNSTs	Further study needed	-	[124]
Torkinib	mTOR	MPNSTs	Further study needed	-	[113]
Trabectedin	Alkylating Agent	MPNSTs	Further study needed	-	[127]
Trametinib	MEK	pNFs, MPNSTs	Phase II Study (NCT03741101 -Active, not recruiting)	Monotherapy	[68,104,123,136]
Tranilast	NLRP3, TGFB, MAPK	pNFs, MPNSTs	Further study needed	-	[121]
Triciribine phosphate	AKT1	pNFs	Further study needed	-	[104]
UC1	NAB3; Proposed	MPNSTs	Further study needed	-	[54]
UNC2250	MER	MPNSTs	Further study needed	-	[127]
Vandetanib	RET, EGFR, VEGFR	MPNSTs	Further study needed	-	[119]
Varlitinib tosylate	EGFR	pNFs	Further study needed	-	[104]
Verteporfin	YAP	MPNSTs	Further study needed	-	[60]
Vincristine	TUBB, TUBA4A	pNFs, LGGs, MPNSTs	Phase II Study (NCT00846430- Completed); Phase III Study (NCT03871257- Active, not recruiting); Phase III Study (NCT04166409-Recruiting)	Combinatorial	[119]
Volasertib/BI6727	PLK1	Melanoma ^#^, MPNSTs	Further study needed	-	[116,117]
Vorinostat	HDAC	MPNSTs	Further study needed	-	[49,119,127,140]
Y100/Y100B	Unknown	GBMs	Further study needed	-	[130]
Y102	BORC; Proposed	GBMs, pNFs, MPNSTs	Further study needed	-	[131]

^#^ Tested on a cell line originally misidentified as an MPNST.

## 9. Discussion and Future Directions

Drug screening efforts for NF1-associated tumors have advanced considerably; however, significant challenges remain, especially in translating preclinical findings into effective therapies for malignant and heterogeneous tumors such as MPNSTs. The continued innovation in developing model systems, combination therapy strategies, and personalized approaches will be essential to further advance NF1 tumor therapeutics.

### 9.1. MEK Inhibitor Resistance Screens

Resistance to MEK inhibitors has been reported in both pNFs and MPNSTs, typically mediated via adaptive feedback pathways and compensatory signaling pathways, such as PI3K/mTOR [162]. Future screening to reveal the resistance pathways could potentially reveal susceptibilities that can be targeted either by vertical inhibition (inhibition of multiple nodes within one pathway) or horizontal combination therapy in convergent pathways.

### 9.2. Alternative High-Throughput Screening (HTS) Models for Drug Discovery

Organoid cultures and tumor-on-a-chip models are new technologies with capacity to recapitulate the tumor microenvironment and provide novel platforms for personalized NF1 drug screening [48,163]. Furthermore, the improvement in single-cell and spatial omics approaches will provide new information on drug resistance mechanisms and tumor heterogeneity. Although most existing HTS methods record cell viability or growth, other phenotypic reads (e.g., cell morphology profiling through techniques like Cell Painting) could yield richer, multidimensional data to guide drug discovery [164].

### 9.3. Targeting the Tumor Microenvironment

There are considerable efforts underway to better characterize the immune and stromal landscape of these tumors, particularly in MPNSTs [165,166]. These include discovering druggable targets in the tumor microenvironment (TME) and developing strategies to modulate immune responses, including through immune checkpoint inhibitors. A deeper understanding of the TME will be essential for the success of future immunotherapeutic interventions.

### 9.4. The Role of Artificial Intelligence in Drug Discovery

Artificial intelligence (AI) has the power to transform drug discovery by enabling the integration of high-dimensional data (such as transcriptomics and proteomics) to predict drug sensitivity, repurposing discovery, and rational combination therapy design. Virtual screening has not only hastened drug development, but it is also time- and cost-saving by enabling prioritization of candidates in a streamlined manner [167]. AI provides numerous applications that can be harnessed for drug discovery and NF1 development.

One of the principal applications of AI is computer-aided drug design (CADD), in which AI software is able to screen swiftly through huge chemical libraries and model interactions with NF1-relevant targets—like MEK, mTOR, or RAS pathway members—thereby accelerating the identification of promising therapeutic candidates. Furthermore, binding affinity prediction using machine learning models trained on structural and biochemical data may help prioritize compounds with high potential to engage molecular targets implicated in NF1 tumor growth, including those considered “undruggable” by traditional standards (see below). AI will also likely be able to assist in streamlining the repurposing of existing drugs for NF1 by analyzing complex patterns in biomedical data to reveal novel disease-drug associations. This can help prioritize agents for screening in high-throughput assays in the lab. Patient-specific toxicity prediction could benefit from the AI integration of multi-omic data sets (genomics, transcriptomics, proteomics) and patient electronic health records for predicting adverse drug reactions. This is particularly valuable in NF1, given the heterogeneity of tumor burden, mutation profile, and therapeutic responses.

The emerging AI-enabled strategies offer powerful tools to enhance the precision, efficiency, and success rate of NF1 drug discovery, from initial screening to clinical translation. The first AI-born treatment for NF1, HLX-1502, is advancing toward a Phase II clinical trial for patients with pNFs. Developed by Healx on an in-house AI platform and tested in a well-established *in vivo* preclinical model, HLX-1502 exemplifies the potential of AI-driven drug development [168,169].

### 9.5. Addressing “Undruggable” Targets

Historically, “undruggable” proteins have posed significant challenges due to their structural features, lack of well-defined binding pockets, and essential cellular roles. These include small GTPases, such as the RAS family. Despite some advances in KRAS-targeted therapies [170], these approaches have remained limited in scope and are often hindered by rapid development of resistance.

Recent advances in protein structure prediction, most notably through AlphaFold, have revolutionized the study of protein architecture and dynamics [171]. AlphaFold enables high-accuracy prediction of three-dimensional (3D) protein structures directly from amino acid sequences, including for proteins that previously lacked experimental structural data. These structural predictions open new avenues in drug discovery. These include in silico identification of cryptic binding pockets, offering novel opportunities for allosteric modulation or fragment-based screening, particularly in proteins previously deemed undruggable. Accurate 3D structures also permit structure-based drug design (SBDD) principles to model interactions between proteins and small molecules.

Beyond the direct targeting of undruggable proteins, alternative strategies include modulating their expression or function through upstream regulators or epigenetic mechanisms. AI platforms can also assist in identifying synthetic lethal interactions or network-based vulnerabilities that bypass the need for direct inhibition.

## 10. Conclusions

NF1 presents unique, complex challenges in tumor management due to its diverse tumor types and underlying molecular heterogeneity. The powerful tools of high-throughput and targeted drug screening have identified novel therapeutic candidates and synergistic drug combinations. These screening approaches have enabled the rapid evaluation of large compound libraries and genetic vulnerabilities specific to NF1-associated tumors. While promising leads have been identified, translating these findings into effective clinical therapies will require further validation and optimization in preclinical animal models. We are confident that the continued integration of advanced screening technologies with new molecular insights holds great potential to improve outcomes for patients with NF1-related tumors.

**Table 2 curroncol-32-00649-t002:** Comprehensive list of cell and animal models used in screens referenced in this review.

Name	Source/Model	*NF1* Mutation(s) (If Known)	*NF1* Status	Used in
** *iPSCs and derived cell lines* **				
Patient #2 NF1^+/−^ iPSC	iPSC derived from peripheral blood mononuclear cells from NF1 patient	c.3431_3432dupGT	*NF1^+/^* ^−^	[47]
Patient #2 NF1^−/−^ iPSC	Patient #2 *NF1^+/^*^−^ iPSC with CRISPR KO of remaining wild-type *NF1* allele	c.3431_3432dupGT; LOH	*NF1* ^−/−^	[47]
WTC-mEGFP-Safe harbor locus (AAVS1)-cl6 (RRID:CVCL_JM19)	iPSC from healthy individual with eGFP inserted at the safe harbor locus AAVS1 under CAGGS (Coriell Institute (NJ, USA)	n/a	*WT*	[48]
mEGFP-PT	iPSC mEGFP with CRISPR *PTEN* and *TP53* loss	n/a	*WT*	[48]
mEGFP PTCC	iPSC mEGFP iPSC with CRISPR *PTEN*, *TP53*, and *CDKN2A/B* loss	n/a	*WT*	[48]
mEGFP PTN	iPSC mEGFP iPSC with CRISPR *PTEN*, *TP53*, and *NF1* loss	*NF1* CRISPR KO	*NF1^−/−^*	[48]
C6-a (RUID: 06C53141)	iPSC from healthy individual—RUCDR Infinite Biologics	n/a	*WT*	[48]
C6-a PT	C6-a with CRISPR *PTEN* and *TP53* loss	n/a	*WT*	[48]
C6-a PTCC	C6-a with CRISPR *PTEN*, *TP53* and *CDKN2A/B* loss	n/a	*WT*	[48]
C6-a PTN	C6-a with CRISPR *PTEN*, *TP53*, and *NF1* loss	*NF1* CRISPR KO	*NF1^−/−^*	[48]
** *Organoids derived from cutaneous neurofibroma (cNF) NF1 patient samples* **				
NF0002-7	cNF	*NF1* deletion (by whole genome sequencing)	*NF1^−/−^*	[61]
NF0004	cNF	No sequencing data available	*NF1^−/−^*	[61]
NF0004-6_8	cNF	No sequencing data available	*NF1^−/−^*	[61]
NF0009	cNF	*NF1* inversion (by whole genome sequencing)	*NF1^−/−^*	[61]
NF00012	cNF	*NF1* deletion, inversion, and stop gain (by whole genome sequencing)	*NF1^−/−^*	[61]
** *Epithelial-like and fibroblast cell lines* **				
HSC1	Epithelial-like primary culture isolated from human spinal nerves	n/a	*WT*	[116]
HSC2	Primary culture isolated from human spinal nerves	n/a	*WT*	[116]
HFF	Fibroblasts	n/a	*WT*	[42]
** *Immortalized Schwann cell (SC) lines including those derived from plexiform neurofibromas (pNFs)* **				
ipn02.3 2λ	Normal SC from healthy individual	n/a	*WT*	[41,50,102,103,111,129]
ipn02.3 2λ C8	CRISPR-edited ipn02.3 2λ	Exon 3 (1 bp del)	*NF1^+/−^*	[50]
ipn02.3 2λ C23	CRISPR-edited ipn02.3 2λ	Exon 3 (2 bp del), Exon 3 (1 bp del)	*NF1^−/−^*	[50]
ipn02.8	Normal SC from healthy individual	n/a	*WT*	[41,102,103]
ipn97.4	Normal SC from healthy individual	n/a	*WT*	[41,103]
ipNF00.6	pNF	Gene deletion (>1 Mb); Unknown	*NF1^−/−^*	[41]
ipNF03.3	pNF	c.4269 G > A in-frame exon skip; Unknown	*NF1^−/−^*	[41]
ipNF04.4	pNF	R2237X; LOH	*NF1^−/−^*	[41,102]
sipnNF95.12B	pNF	L216P	*NF1^+/−^*	[41,102]
ipNF05.5	pNF	c.3456_3457insA; LOH	*NF1^−/−^*	[41,50,102,111,129]
ipNF05.5-MX (six clone mix)	pNF	c.3456_3457insA; LOH	*NF1^−/−^*	[41,102,104]
ipNF06.2A	pNF	G848W; Unknown	*NF1^−/−^*	[41,102,104]
ipNF95.11b C	pNF	c.1756delACTA; LOH	*NF1^−/−^*	[41,50,102,103,129,131]
ipNF95.11b C/T	pNF	c.1756delACTA; LOH	*NF1^−/−^*	[41,103,104]
ipNF95.6	pNF	R816X; R2237X	*NF1^−/−^*	[41,102,103,104,111,129]
ipnNF09.4	pNF	c.3456_3457insA	*NF1^+/−^*	[41,50]
ipnNF95.11C	pNF	c.1756delACTA	*NF1^+/−^*	[41,50,102,104,131]
HSC1λ	Normal SC—ipn02.3 2λ	n/a	*WT*	[49,60]
HSC1λ N0 (5)	CRISPR-derived from HSC1λ line	*NF1* indels in exon 10 on both alleles	*NF1^−/−^*	[49,60]
HSC1λ N1 (10)	Went through the CRISPR process—unedited	n/a	*WT*	[49,60]
** *Malignant peripheral nerve sheath tumor (MPNST) cell line models* **				
ST88-3/88-3/NF88-3	MPNST	c.6952T > C; LOH	*NF1^−/−^*	[78,113,117,118]
NF90-8/90-8/90.8/90-8TL	MPNST	p.Asp1302Tyrfs*5 (c.3904_3910del); LOH	*NF1^−/−^*	[69,78,84,111,113,117,118,125]
ST88-14/ST8814 ^☨^	MPNST	p.Arg304Ter (c.910C > T); LOH	*NF1^−/−^*	[50,69,78,84,87,106,109,111,113,116,117,118,125,128]
T265 ^^^	MPNST	p.Arg304Ter (c.910C > T); LOH	*NF1^−/−^*	[54,113,117,128]
S1844.1	MPNST	LOH in both alleles	*NF1^−/−^*	[109]
S1507.2/S1507-2	MPNST	Splicing mutation in intron 23-1; deletion in exon 10a	*NF1^−/−^*	[109,116]
S462	MPNST	c.6792C > A; LOH	*NF1^−/−^*	[60,69,87,109,111,116,124,125,128,129]
YST-1	Sporadic MPNST	n/a	*WT*	[116]
sNF94.3	MPNST	Microdeletion	*NF1^+/−^*	[118,123]
SNF10.1	MPNST	R1276X; Hemizygous deletion	*NF1^−/−^*	[123]
sNF02.2	MPNST	c.4868A > T	*NF1^+/−^*	[113,118,123]
sNF96.2	MPNST	p.Asn1229Metfs*1; LOH	*NF1^−/−^*	[69,113,118,121,123,124,125,131]
HS-PSS	Sporadic MPNST	n/a	*WT*	[116,124]
S462.TY/S462 TY	MPNST	p.Tyr2285Ter (c.6855C > A); p.Tyr2264Ter, (c.6792C > A)	*NF1^−/−^*	[49,85,113,117,118,128]
NCC-MPNST1-C1	MPNST	Mutation not detected	*NF1^−/−^* (Presumed)	[119,127]
NCC-MPNST2-C1	Sporadic MPNST	p.Leu179Tyrfs*11 (c.536_539del)	*NF1^−/−^* (Presumed)	[119,127]
NCC-MPNST3-C1	MPNST	p.Arg816Ter (c.2446C > T)	*NF1^−/−^* (Presumed)	[119,127]
NCC-MPNST3-X2-C1	From 2nd generation xenograft of MPNST	p.Arg816Ter (c.2446C > T)	*NF1^−/−^* (Presumed)	[119]
NCC-MPNST4-C1	Sporadic MPNST	c.5812 + 3delAGTA	*NF1^−/−^* (Presumed)	[119,127]
NCC-MPNST5-C1	MPNST	p.Thr586Valfs*18 (c.1756_1759delACTA)	*NF1^−/−^* (Presumed)	[119,127]
NCC-MPNST6-C1	Sporadic MPNST	Not Reported	*NF1^+/+^* (Presumed)	[127]
NMS-2	MPNST	c.7062 + 1G > T (c.6999 + 1G > T); LOH	*NF1^−/−^*	[125]
NF1-08	MPNST	c.701_730 + 10del; LOH	*NF1^−/−^*	[125]
NF1-09	MPNST	c.6792C > A p.(Tyr2264*); c.1186-5_1186-1del	*NF1^−/−^*	[125]
NF1-18B	MPNST	c.1642-449 > G; LOH	*NF1^−/−^*	[125]
SP-10	Sporadic MPNST	g.30922951_31318216del	*NF1^−/−^*	[125]
** *Patient-derived MPNST xenograft cell line models* **				
JH-2-002	PDX	Microdeletion; c.6308T > C and c.6309_6310del	*NF1^−/−^*	[87]
JH-2-031	PDX	Microdeletion; c.4771dup	*NF1^−/−^*	[87]
JH-2-079	PDX	c.5812 + 1G > A; p.C2223X (c.6669C > A)	*NF1^−/−^*	[87]
NF1-18B	PDOX	c.1642-449 > G; LOH	*NF1^−/−^*	[125]
SP-10	PDOX	g.30922951_31318216del	*NF1^−/−^*	[125]
** *Melanoma ^#^* **				
STS-26T	Melanoma ^#^	n/a	*WT*	[54,69,86,113,116,117,123,124,128]
HS-Sch-2	Melanoma ^#^	c.3113 + 1G > A; p.Glu91Asnfs*6 (c.270_288del)	*NF1^−/−^*	[116,124]
** *High-grade glioma (HGG) cell line models* **				
JHH-NF1-GBM1	NF1-associated Human HGG	Loss of function of NF1	No detectable protein by western	[142]
TM-31	Human HGG	pLF1247fs*18, homozygous	*NF1^−/−^*	[140]
LN319	Human HGG	Total inactivation	*NF1^−/−^*	[140]
NF1-HGG 17	HGG from *NPcis (Nf1^+/−^ and Tp53^+/−^ in cis) *mouse	Complete *Nf1* loss	*Nf1^−/−^*	[140]
NF1-HGG 5653	HGG from *NPcis (Nf1^+/−^ and Tp53^+/−^ in cis) *mouse	Complete *Nf1* loss	*Nf1^−/−^*	[140]
NF1-HGG 5746	HGG from *NPcis (Nf1^+/−^ and Tp53^+/−^ in cis) *mouse	Complete *Nf1* loss	*Nf1^−/−^*	[140]
** *Glioblastoma (GBM) cell line models* **				
BR 23C	GBM ^=^	I526S	*NF1^−/−^* (Presumed)	[136]
HSR-GBM1	GBM ^=^	A1676T	*NF1^+/−^*	[136]
JHH-68	Non-NF1 GBM	n/a	*WT*	[136]
JHH-136	Non-NF1 GBM	n/a	*WT*	[136]
JHH-227	Non-NF1 GBM	n/a	*WT*	[136]
JHH-505	Non-NF1 GBM	n/a	*WT*	[136]
JHH-520	GBM ^=^	Homozygous deletion	*NF1^−/−^*	[136]
JHU-0879	Non-NF1 GBM	n/a	*WT*	[136]
JHU-1016B	GBM ^=^	L115T fs*42	*NF1^+/−^*	[136]
GBM43	GBM ^=^	Inactivating mutation; no detectable protein by western	*NF1^−/−^*	[137]
SB28	C57BL/6 Mouse GBM	n/a	*WT*	[137]
U251-MG	Non-NF1 GBM	c.2033dupC; no *WT* allele present	*NF1^−/−^*	[130,131,142]
U251 ATRX^−/−^	Non-NF1 GBM; CRISPR modified for *ATRX* deletion	c.2033dupC; no *WT* allele present	*NF1^−/−^*	[142]
U251 ATRX^−/2.02^	Non-NF1 GBM; CRISPR modified for *ATRX* deletion	c.2033dupC; no *WT* allele present	*NF1^−/−^*	[142]
U87-MG	Non-NF1 GBM	Proteasome-mediated degradation of NF1	No detectable protein by western	[130,131]
LN229	GBM ^=^	Total gene deletion	*NF1^−/−^*	[87]
U373	Non-NF1 GBM	Unknown	No detectable protein by western	[87]
** *Mouse Embryonic Fibroblast (MEF) cell line models* **				
*Nf1wt*	MEFs derived from *Nf1wt E1A-p53* mouse	n/a	*WT*	[118]
*Nf1* * ^−/−^ *	MEFs derived from *Nf1^−/−^ E1A-p53* mouse	Complete *Nf1* loss	*Nf1^−/−^*	[118]
** *Mouse astrocytoma cell line models* **				
*Nf1*^GFAP^CKO	Astrocytes from *Nf1*^GFAP^CKO mouse	*Nf1^−/−^* astrocytes only	*Nf1^−/−^*	[106]
K5001	Spontaneous astrocytoma from *NPcis (Nf1^+/−^ and Tp53^+/−^ in cis) *mouse	Mutant *Nf1* allele; loss of *WT* allele	*Nf1^−/−^*	[134]
KR158	Spontaneous astrocytoma from *NPcis (Nf1^+/−^ and Tp53^+/−^ in cis) *mouse	Mutant *Nf1* allele; loss of *WT* allele	*Nf1^−/−^*	[134]
K1492	Spontaneous astrocytoma from *NPcis (Nf1^+/−^ and Tp53^+/−^ in cis) *mouse	Mutant *Nf1* allele; loss of *WT* allele	*Nf1^−/−^*	[134]
** *Mouse models* **				
*Dhh*-Cre; *Nf1*fl/fl	Mouse	Cre recombinase removes both *Nf1* alleles in Schwann cells	*Nf1^−/−^*	[110]
*LSL-Trp53^R270H^* *(Cnp-hEGFR background)*	Mouse	Conditional mutant alleles of *p53* using *loxP-STOP-loxP*	*WT*	[128]
**Saccharomyces cerevisiae *(Budding yeast) models***				
MLY41a	Yeast	n/a	*WT*	[54,130,131]
MDW057	Yeast	n/a	*WT*	[54,130,131]
MDW028	Yeast	*ira2Δ*	*ira2^−/−^*	[54,130,131]
MDW035	Yeast	*erg6Δira2Δ*	*ira2^−/−^*	[54,130,131]
**Drosophila melanogaster *(Fruit fly) models***				
S2R+	*Drosophila* Schneider 2 cells	n/a	*WT*	[50]
S2R+ NF1-KO ^&^	CRISPR-derived *Drosophila* Schneider 2 cells	del c.148–161, del c.150–160, del c.148–160	*dNf1^−/−^*	[50]
*Nf1^C1^*	*dNf1*-deficient *Drosophila* model	del c.160–161AT	*dNf1^−/−^*	[50]
**Danio rerio *(Zebrafish) models***				
*nf1a^+/−^*; *nf1b^−/−^*; *p53^m/m^*; *sox10:mCherry*	Zebrafish	Homozygous loss of *nf1b* and heterozygous for *nf1a*	*nf1a^+/−^*; *nf1b^−/−^*	[69]

^#^ Originally misidentified as an MPNST. ^^^ Now thought to be a derivative of ST88-14. ^☨^ Recurrence of ST88-3. ^=^ Unclear if donor was an NF1 patient. ^&^ Triploid for *dNf1*.

## Figures and Tables

**Figure 1 curroncol-32-00649-f001:**
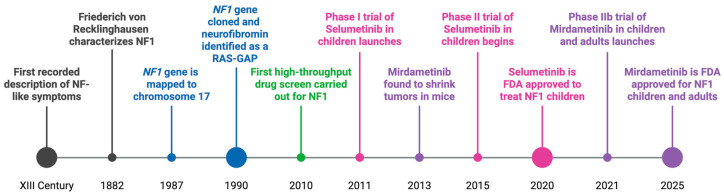
Timeline of critical NF1 discoveries. Although neurofibromatosis-like symptoms were first documented centuries ago, the most significant discoveries related to NF1 have occurred within the last 35 years. Figure based on information obtained from [23,24,25,26,27,28,29,30,31,32,33].

**Figure 2 curroncol-32-00649-f002:**
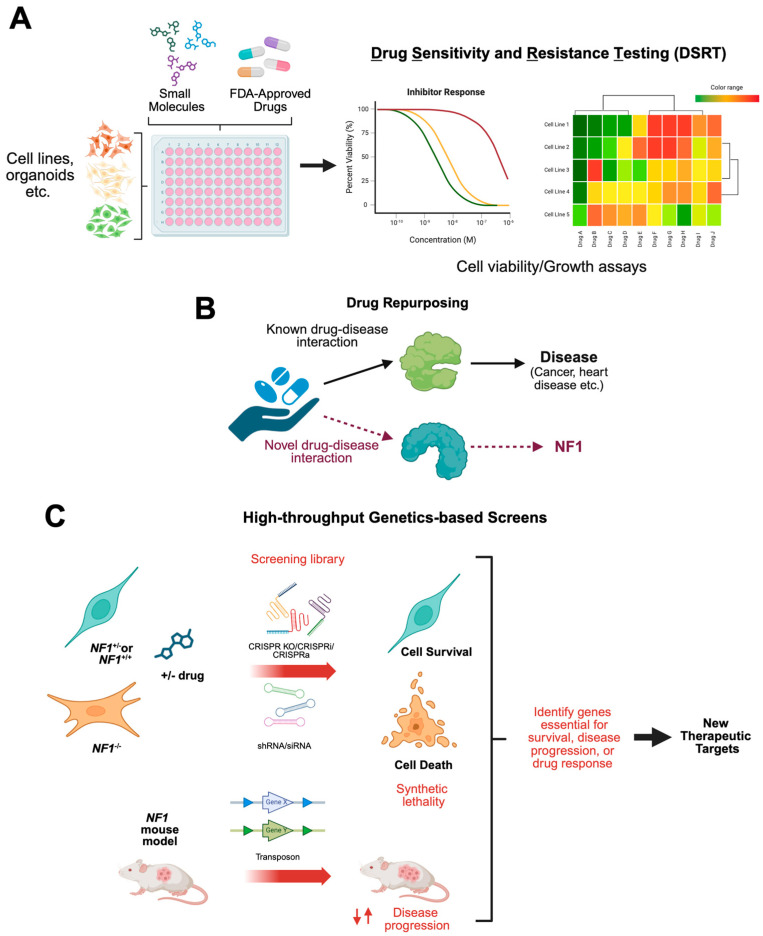
High-throughput screens for drug discovery and therapeutic target identification. High-throughput screens (HTS) are powerful tools used in drug discovery and functional genomics to rapidly assess the biological activity of large numbers of compounds or genetic perturbations. (**A**) Drug sensitivity and resistance testing (DSRT) is a high-throughput pharmacological screening method, where libraries of small molecules are tested on wild-type (*WT*) control cells and cells harboring disease-relevant genotypes to identify potential drug candidates that selectively affect diseased cells while preserving WT cell viability. In the context of NF1, ideal compounds reduce the viability of *NF1*-deficient tumor cells but spare normal *NF1*-heterozygous cells from toxicity. (**B**) Drug repurposing: This strategy is particularly valuable for rare diseases like NF1 because it accelerates FDA approval by leveraging drugs with established mechanisms of action, bypassing the need to develop novel, uncharacterized compounds from scratch. (**C**) Genetic-based high-throughput screens can be used to systematically perturb genes to identify those essential for survival, disease progression, or drug response. They include CRISPR/Cas9 knockout (CRISPR KO), CRISPR interference/activation (CRISPRi/a) and RNAi (shRNA, siRNA) screens. Synthetic lethality occurs when the simultaneous loss of two genes results in cell death, whereas the loss of either gene alone is non-lethal. Transposon screens in mouse models of NF1 have been used to validate and identify new genetic drivers of tumor progression. Figure based on information obtained from [34,36,37,38].

**Figure 3 curroncol-32-00649-f003:**
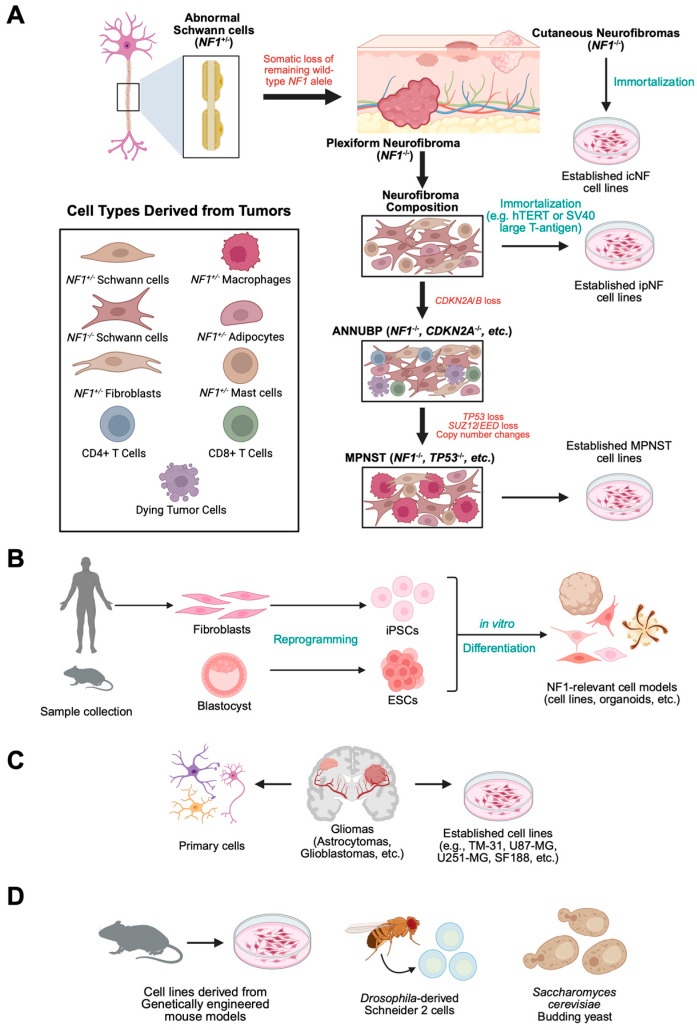
*In vitro* cell models of NF1-associated tumors. (**A**) Cutaneous neurofibromas (cNFs) originate from *NF1* homozygous (*NF1^−/−^*) Schwann cells and are confined to the skin or just beneath it. Plexiform neurofibromas (pNFs) arise from *NF1* heterozygous (*NF1^+/−^*) Schwann cells following somatic loss of the remaining wild-type *NF1* allele. These tumors recruit additional cell types, including fibroblasts and mast cells, which contribute to tumor progression. pNFs typically develop deep along major nerve trunks but can also affect nerves closer to the skin. Benign pNFs can undergo malignant transformation into malignant peripheral nerve sheath tumors (MPNSTs) through intermediate lesions known as atypical neurofibromas or atypical neurofibromatous neoplasms of uncertain biological potential (ANNUBPs). While pNF tumorigenesis is driven solely by biallelic *NF1* loss, their progression from pNFs to ANNUBPs and ultimately to MPNSTs involves the accumulation of additional genetic alterations, such as mutations in tumor suppressor genes like *CDKN2A/B* and *TP53*, *EED/SUZ12*, driving tumor aggressiveness and malignancy. cNF and pNF cells from patient biopsies have been immortalized (icNFs, ipNFs) using various methods, such as hTERT overexpression or SV40 large T antigen transfection, to ensure access to relevant cell models for extended periods of time. In contrast, MPNST cell lines have been established without the need for immortalization. (**B**) Stem cells derived from humans or animal models of NF1 serve as versatile tools for drug screening. Human stem cells are primarily generated by reprogramming fibroblasts into induced pluripotent stem cells (iPSCs), while embryonic stem cells (ESCs) can be derived from blastocysts. These stem cells can be further differentiated into NF1-relevant cell types for use in both 2D cultures and 3D organoid models. (**C**) Additional cell sources include glioma biopsies, which can yield primary cultures or be immortalized to establish stable cell lines for extended studies. (**D**) *NF1*-deficient cell lines derived from animal models of NF1 such as mice and *Drosophila*, and single-celled model organisms such as budding yeast (*Saccharomyces cerevisiae*), have also been employed to investigate drug mechanisms and molecular pathways affected by *NF1* loss using high-throughput screens. Figure created from information derived from [4,5,6,7,8,9,10,11,12,38,44,45,47,48,49,50,51,52,53,54,55,56,57].

**Figure 4 curroncol-32-00649-f004:**
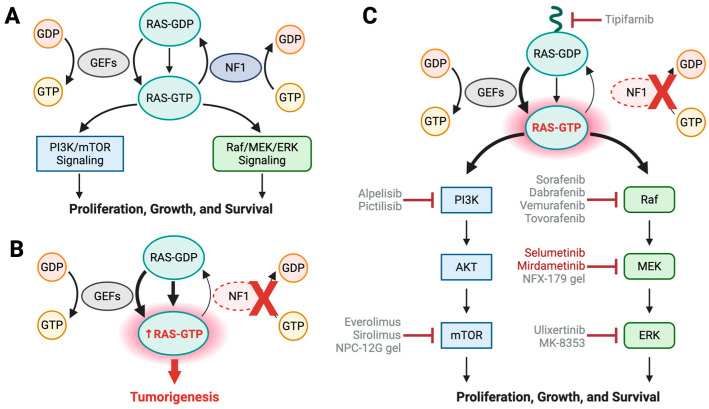
Neurofibromin is a critical regulator of RAS activity. (**A**) Under normal conditions, RAS is activated by RAS-GEFs, which promote downstream PI3K/mTOR and Raf/MEK/ERK signaling, regulating normal cell proliferation, growth, and survival. When these signals need to be reduced, neurofibromin, functioning as a RAS-GAP, stimulates the intrinsic GTP hydrolysis in RAS to convert it to its inactive form. (**B**) When neurofibromin is lost, the intrinsic GTP hydrolysis of RAS is too slow to inactivate RAS, forcing a shift to active, GTP-bound RAS, as RAS-GEFs are still functional. This leads to increased signaling through downstream RAS pathways and tumorigenesis. (**C**) Strategies to reduce hyperactive RAS signaling caused by *NF1* loss: several small-molecule inhibitors have been tested, with targets located downstream of RAS, as well as against RAS modifiers themselves. The only successful inhibitors to receive FDA approval to date to treat NF1 tumors are selumetinib and mirdametinib (red). Figure based on information obtained from [72,73,74,75,76,77].

## Data Availability

No new data were created or analyzed in this study.

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
