# Peer review of "Neurofibromatosis Type 1 and the Search for Effective Tumor Therapies Using High-Throughput Drug Screening"

_curroncol, 2025, doi:10.3390/curroncol32110649_

Round 1
Reviewer 1 Report
Comments and Suggestions for Authors
The manuscript by Bouley et al is excellent overall. It is an original scholarly contribution to the field and does not duplicate work that is already available. The English is clear. The attention to detail is high and only a single minor typo was found:
- Typo on line 505, no "T" in there.
Other things that could be considered to further improve the work:
On line 853 they mention "an NF1 patient with a sporadic HGG", which could be confusing - can there be a "sporadic" tumor in an NF1 patient when they are at higher risk of developing an HGG due to the underlying condition? On line 894-5, they claim mTOR inhibition was identified as a vulnerability in MPNSTs through HTS. Several non-HTS studies identified and tested mTOR inhibitors and pre-date the cited study.
-
- PMID: 19634141
- PMID: 18164202
- PMID: 18483311
The final suggestion is that it would improve the review to include a summary data table of the drugs identified through HTS, to emphasize those that still have potential to develop into therapeutics. The table already included is great to summarize as models used. However, the goal of HTS is drug identification, so it seems to miss an opportunity to summarize only the models but not the drug findings. For example, digoxin was found to have efficacy in MPNST and cNF, but no further data seem to have been produced, while PLK1 inhibitors have efficacy in MPNSTs and could be investigated in combination and not only as monotherapies. It would be useful to summarize for the reader whether the drugs are in clinical trials, failed a clinical trial, need more research, could be promising in combination therapies etc.
Author Response
Comment 1: Typo on line 505, no "T" in there.
Response 1: Line 505 “T” added (typo)
Comment 2: On line 853 they mention "an NF1 patient with a sporadic HGG", which could be confusing - can there be a "sporadic" tumor in an NF1 patient when they are at higher risk of developing an HGG due to the underlying condition?
Response 2: Lines 852-854 Replaced with: The cell lines used were derived from either an NF1 patient with a malignant astrocytoma or Nf1-deficient mice that spontaneously developed HGGs [140,141]. We are grateful for the reviewer bringing the original ambiguous sentence to our attention. We had meant to say that the mouse developed sporadic HGGs, not the patient. We also added the citation [140] to Dougherty et al (2023) which was missing.
Comment 3: On line 894-5, they claim mTOR inhibition was identified as a vulnerability in MPNSTs through HTS. Several non-HTS studies identified and tested mTOR inhibitors and pre-date the cited study. PMID: 19634141, PMID: 18164202, and PMID: 18483311
Response 3: Lines 894 to 897: Replaced with: For example, while mTOR inhibition was identified as a potential therapeutic vulnerability in MPNSTs through HTS [106] as well as multiple non-HTS studies [143–145], testing of mTOR inhibitors such as everolimus and sirolimus (also known as rapamycin) have shown limited efficacy as single agents in pNFs and cNFs [99,146–148]. We appreciate comment from Reviewer 1 suggesting to cite three references for non-HTS studies which identified and tested mTOR inhibitors.
Comment 4: he final suggestion is that it would improve the review to include a summary data table of the drugs identified through HTS, to emphasize those that still have potential to develop into therapeutics. The table already included is great to summarize as models used. However, the goal of HTS is drug identification, so it seems to miss an opportunity to summarize only the models but not the drug findings. For example, digoxin was found to have efficacy in MPNST and cNF, but no further data seem to have been produced, while PLK1 inhibitors have efficacy in MPNSTs and could be investigated in combination and not only as monotherapies. It would be useful to summarize for the reader whether the drugs are in clinical trials, failed a clinical trial, need more research, could be promising in combination therapies etc.
Response 4: Line 941; Addition of new table (Table 1: Status of drugs identified through high-throughput screening in NF1). Line 1029; Table 1 now becomes Table 2
Reviewer 2 Report
Comments and Suggestions for Authors
This comprehensive review on Neurofibromatosis Type 1 and the Search for Effective Tumor Therapies using High-Throughput Drug Screening is well-written and gives a thorough overview of the relevant literature. Clear and well-designed figures accompany the text, and the Table contains a practical list of cell and animal models used in screens for NF1 therapies.
Author Response
We thank reviewer 2 for their time and energy in reviewing our manuscript, and we appreciate their feedback.
Reviewer 3 Report
Comments and Suggestions for Authors
author's manuscript includes a comprehensive review of pre-clinical and translational knowledge on NF1-associated tumors attempting at narrowing the gap between in vitro and in vivo studies and patients' management and lay foundation for novel translational approaches to test new drugs and/or combinatorial strategies in NF1-associated tumors, highlighting the central role of MAPK/MEK pathway
the manuscript is acceptable for publication in its present form without major reviews
Author Response
We thank reviewer 3 for their time and energy in reviewing our manuscript, and we appreciate their support.